# Mixture Compressor for Mixture-of-Experts LLMs Gains More

**Wei Huang**[* 1], **Yue Liao**[* 2 4], **Jianhui Liu**[1], **Ruifei He**[1], **Haoru Tan**[1]
**Shiming Zhang**[† 1], **Hongsheng Li**[2 4], **Si Liu**[† 3], **Xiaojuan Qi**[† 1]
[1]The University of Hong Kong  [2]The Chinese University of Hong Kong  [3]Beihang University
[4]Centre for Perceptual and Interactive Intelligence, Hong Kong

## Abstract

Mixture-of-Experts large language models (MoE-LLMs) marks a significant step forward of language models, however, they encounter two critical challenges in practice: **1)** expert parameters lead to considerable memory consumption and loading latency; and **2)** the current activated experts are redundant, as many tokens may only require a single expert. Motivated by these issues, we investigate the MoE-LLMs and make two key observations: **a)** different experts exhibit varying behaviors on activation reconstruction error, routing scores, and activated frequencies, highlighting their differing importance, and **b)** not all tokens are equally important– only a small subset is critical. Building on these insights, we propose **MC**, a training-free **M**ixture-**C**ompressor for MoE-LLMs, which leverages the significance of both experts and tokens to achieve an extreme compression. First, to mitigate storage and loading overheads, we introduce *Pre-Loading Mixed-Precision Quantization (PMQ)*, which formulates the adaptive bit-width allocation as a Linear Programming (LP) problem, where the objective function balances multi-factors reflecting the importance of each expert. Additionally, we develop *Online Dynamic Pruning (ODP)*, which identifies important tokens to retain and dynamically select activated experts for other tokens during inference to optimize efficiency while maintaining performance. Our **MC** integrates static quantization and dynamic pruning to collaboratively achieve extreme compression for MoE-LLMs with less accuracy loss, ensuring an optimal trade-off between performance and efficiency. Extensive experiments confirm the effectiveness of our approach. For instance, at 2.54 bits, MC compresses 76.6% of the model, with only a 3.8% average accuracy loss in eight commonsense benchmarks. During dynamic inference, we further reduce activated parameters by 15%, with a performance drop of less than 0.6%. Remarkably, MC even surpasses floating-point 13b dense LLMs with significantly smaller parameter sizes, suggesting that mixture compression in MoE-LLMs has the potential to outperform both comparable and larger dense LLMs. Our code is available at https://github.com/Aaronhuang-778/MC-MoE.

## 1 Introduction

Mixture-of-Experts large language models (MoE-LLMs) (Muennighoff et al., 2024; Jiang et al., 2024; Dai et al., 2024) provide an efficient model-scaling mechanism by utilizing a sparse architecture, in which only a subset of experts is activated by router. This selective activation boosts computational efficiency and scalability by assigning experts dynamically based on the specific needs of each input. Despite reducing the number of active experts to improve inference efficiency, MoE models still face significant deployment challenges. All experts must be loaded into memory simultaneously, and typically at least two experts are activated during inference, resulting in considerable memory and computational overhead. Even an NVIDIA A100-80GB GPU cannot accommodate typical MoE models like Mixtral 8×7b (Jiang et al., 2024) (Fig. 1(b)). The proposed challenges hinder the deployment of LLM with limited hardware resources which further promotes study on MoE-LLM compression for better deploying model-scaling paradigm.

---

*Equal Contribution. † Corresponding Author

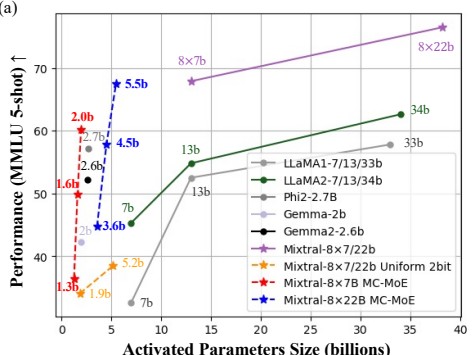 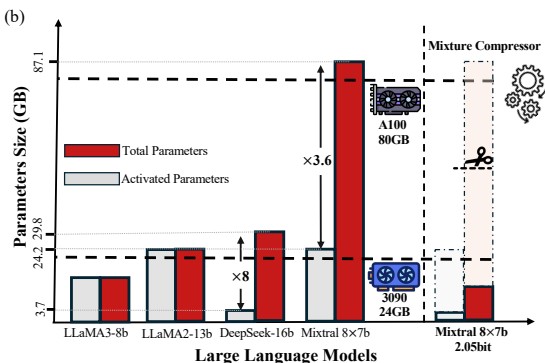

Figure 1: (a) MMLU (5-shot↑) accuracy across different open-source LLMs with various activated parameters (dot-lines denote the quantized models, solid-lines are 16-bit models). To align quantized models' parameter size with 16-bit models, we define 16bits as one parameter (e.g. 8×2-bit elements represent one parameter). (b) Comparison of total parameter size and inference activated parameter size on few open-source LLMs and compressed Mixtral 8×7b.

The primary goal of compressing MoE-LLMs is to reduce the size of expert parameters, as they dominate the memory usage (Li et al., 2024). For instance, in models like Mixtral $8 \times 7b$, the number of expert parameters is 33 times greater than that of the attention modules. On the other hand, recent studies (Chi et al., 2022; Lu et al., 2024) have shown that due to the training strategies of MoE, not all experts are equally important, which indicates that both the static experts during the pre-loading phase and the dynamic experts during online inference need to be compressed. Previous expert compression methods have typically focused on compressing a single phase, such as quantizing expert weights during the pre-loading stage (Li et al., 2024) or pruning experts during the inference stage (Lu et al., 2024; Koishekenov et al., 2022; Kim et al., 2021). Furthermore, vanilla uniform bit-width quantization and expert pruning based solely on routing scores struggle to maintain performance at extremely high compression ratios. Therefore, in this work, we are the first to explore extreme training-free mixture compression for MoE-LLMs, efficiently combining static expert quantization with dynamic expert pruning using a combination of expert importance metrics to achieve ultra-lightweight MoE-LLMs without significantly sacrificing performance.

To this end, we propose the **MC**, *i.e.*, **M**ixture-**C**ompressor for MoE LLMs, exploring the combined benefits of expert quantization and pruning. **MC** consists of two phases: *Pre-Loading Mixed-Precision Quantization (PMQ)* and *Online Dynamic Pruning (ODP)*, as shown in Fig. 1(a). In the pre-loading phase, we focus on extreme compression of the stored experts through low-bit quantization. Our empirical study reveals imbalances in activation reconstruction error, routing weights, and frequencies of activated expert (Sec. 3.2.1 and Fig. 3), which inspires the allocation of different bit-widths to each expert. However, relying solely on the routing frequencies or scores is insufficient to accurately determine the optimal bit-width, as the two distributions may not be consistent but rather the opposite (Li et al., 2024). Therefore, we developed a weighted evaluation function that considers both the frequency and scores of expert activations, as well as the associated quantization loss at different bit-widths. This function is then minimized within a Linear Programming (LP) model to determine the optimal quantization configuration. Utilizing a training-free Post-training Quantization (PTQ) approach, GPTQ (Frantar et al., 2022), *PMQ* achieves high-performance compression at extremely low bit-widths (1.5-bit∼2.5-bit), and our mixed-precision strategy is compatible with other advanced quantization techniques (Tseng et al., 2024; Chen et al., 2024; Shao et al., 2023; Egiazarian et al., 2024; Liao & Monz, 2024). As for inference phase, *ODP* dynamically prunes low-confidence experts for each token based on the routing weights. Our pruning strategy follows two key principles: first, experts with significantly lower routing scores are categorized as "low confidence" and can be pruned (Lu et al., 2024). Second, to prevent attention degradation that solely relies on routing weights, we protect important tokens by considering both attention scores and feature magnitudes. Experiments show that protecting only 2% of the important tokens effectively mitigates pruning loss while maintaining nearly the same compression ratio.

The proposed mixture compression of low bit-width experts improves performance compared to uniform quantized experts or other mixed-precision strategies, even surpassing float-point (FP) models with the same number of activated parameters. Moreover, when compressing Mixtral $8 \times 7b$ to

around 8b (2.54-bit), its activated parameters amounted to only 2b, while even outperforming 16-bit LLaMA2-13b by around 8% on the MMLU (5-shot), as shown in Fig. 1(a). Mixture compression exploits the disparities between MoE experts, for the first time enabling surpassing of smaller FP models of equivalent size under extreme compression without training. This achievement underscores the significant compression potential and practical utility of sparse MoE-LLMs.

## 2 RELATED WORKS

**Mixture-of-Experts LLMs.** LLMs have achieved significant advancements across various natural language domains (Chang et al., 2024; Zhao et al., 2023). Despite their success, these models rely heavily on dense parameters, which presents significant challenges for deployment(Zhou et al., 2024; Zhu et al., 2023). Sparse activated MoE models have been identified as an essential strategy to enhance the cost-performance balance in LLMs. In MoE models, each layer is comprised of several experts, with each token activating only a specific subset, thereby significantly improving efficiency compared to dense models, which activate all parameters for every input (Shazeer et al., 2017; Yun et al., 2024). Recent advancements in LLMs (Brown, 2020) have further popularized MoE-based architectures (Jiang et al., 2024; Muennighoff et al., 2024). Industry-leading models such as Mixtral $8 \times 7$b (Jiang et al., 2024) and Deepseek-R1 (Guo et al., 2025) also incorporate this technology.

**Quantization for LLMs**. Post-Training Quantization (PTQ) is an efficient method that requires no additional training, making it well-suited for large-scale LLMs (Dettmers et al., 2022; Frantar et al., 2022; Xiao et al., 2023; Shao et al., 2023; Lin et al., 2024). Previous studies have investigated the diverse salience of weights and proposed mixed-precision methods to improve low-bitwidth performance by allocating different bitwidths accordingly (Dong et al., 2020; Huang et al., 2024c; Dettmers et al., 2023; Shang et al., 2023; Huang et al., 2024a). Recent research introduced an expert-guided, block-wise mixed-precision benchmark for MoE-LLMs to address the disparities in expert weights (Li et al., 2024); however, developing more effective expert-wise quantization strategies remains a challenge. Codebook-based encoding approaches enable more precise quantization of LLMs and enhance post-quantization performance through fine-tuning (Egiazarian et al., 2024; Tseng et al., 2024). While Quantization Aware Training (QAT) requires significant resources (Chen et al., 2024; Liu et al., 2023b), QAT-based retraining strategies or PTQ combined with additional fine-tuning (Liao & Monz, 2024; Guo et al., 2023; Huang et al., 2024b) are more effective in maintaining the performance of quantized lightweight LLMs.

**Parameter pruning for LLMs**. Parameter pruning is another effective method for neural network compression (Kwon et al., 2022; Hubara et al., 2021), and it has recently become crucial in reducing the size of LLM weights (Frantar & Alistarh, 2023; Sun et al., 2023). Traditional pruning approaches focus on two main techniques: structured and unstructured pruning, both of which selectively zero out certain parameters based on their importance (Zhou et al., 2024). In MoE-LLMs, less important experts can be pruned based on activation frequencies or the statistical characteristics of gating (Kim et al., 2021; Koishekenov et al., 2022; Liu et al., 2024). During the model preloading stage, pruning tends to incur greater loss than quantization at the same compression rate. However, dynamically adjusting the quantization bit-width during inference remains a challenge, whereas pruning offers the flexibility to dynamically select activation parameters during inference (Zhu et al., 2023). Recent work by (Lu et al., 2024) has explored dynamically activating $top\text{-}k$ experts based on gating weights in MoE, significantly improving inference efficiency.

## 3 METHOD

### 3.1 PRELIMINARIES

**Mixture-of-Experts LLM.** In decoder-only MoE-LLMs, conventional feed-forward networks (FFN) are replaced by MoE layer, each having $N$ experts (Gale et al., 2023). The MoE-LLMs selectively activates the $top\text{-}k$ experts for different tokens by a group of routing scores $\mathbf{w}_{top\text{-}k} = \{w_0, w_1, ..., w_{k-1}\}$, $k < N$, generated by a gating layer $\text{G}(\mathbf{t})$. Fig. 2(a) illustrates the experts selection mechanism during the inference phase based on routing scores. Specifically, in the Mixtral $8 \times 7$b model, there are 8 experts and each token $\mathbf{t}$ is routed to the $top\text{-}2$ experts (Jiang et al.,

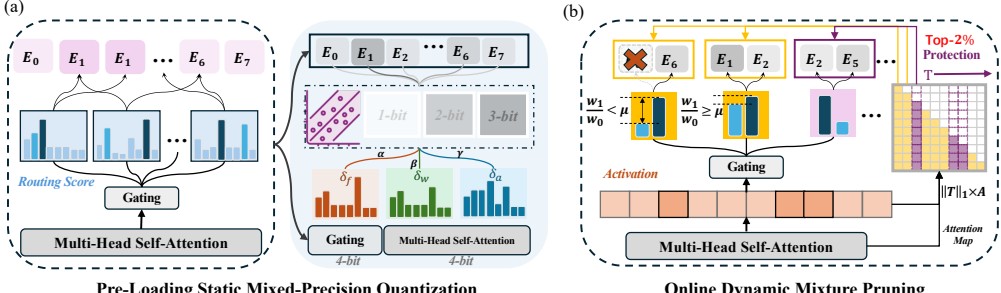

Figure 2: The overview of our proposed MC pipeline with two stages compression for experts. (a) Framework of pre-loading static mixed-precision quantization (PMQ) of MoE-LLMs. PMQ determins the activated feature and loss sensitivity of all experts and plans the optimal precision configuration under ultra-low -bit-width. (b) Schematic of online dynamic mixture pruning (ODP) of MoE-LLMs. ODP utilizes significant token protection mechanism with weigh-guided experts pruning, which only need to keep 2% token to successfully safeguard the MoE performance.

2024). The output $y$ of each token in MoE layer is calculated as:

$$\mathbf{y} = \sum_{w_i \in Top\text{-}2\{G(\mathbf{t})\}} w_i \, E_i(\mathbf{t}), \tag{1}$$

where $E_i$ represents the feed-forward operator of the $i\text{-}th$ expert and $w_i$ is the routing weights/scores calculated by the gating $G(\mathbf{t})$. Therefore, according to the definition in Eq (1), the routing mechanism establishes the correspondence between tokens and experts.

**Quantization Technique.** Since the substantial memory overhead of MoE models mainly arises from the weights of its experts (over 96% weights of the model), quantization is employed for the experts. Specifically, floating-point weights distributed in the interval $[\mathbf{W}_{min}, \mathbf{W}_{max}]$ are mapped to the integer range of $[0, 1..., 2^B]$, where $B$ represents the target bit-width, and the quantization reconstruction for the weights $\mathbf{W} \in R^{in \times out}$ can be defined as:

$$\underset{\mathbf{W}^q}{\arg\min} \ \|\mathbf{W}\mathbf{X} - \mathbf{W}^q\mathbf{X}\|_2^2, \tag{2}$$

where $\mathbf{W}^q$ denotes the quantized weight and $\| \cdot \|_2$ is the mean square error (MSE) loss. The primary objective of this study is to explore the optimal mixture compression strategy for MoE-LLMs. To this end, we employ the efficient PTQ scheme, GPTQ (Frantar et al., 2022), as our foundational tool. By utilizing Hessian-based estimation ($\mathbf{H} = 2\mathbf{X}\mathbf{X}^\top$) and quantization error compensation, GPTQ effectively reduces the group-wise quantization error of weights, enabling the quantization of Mixtral $8 \times 7b$ within 90 minutes. This work focuses on the design of optimal mixture compression strategies for MoE-LLMs and is therefore orthogonal to other quantization techniques, including PTQ methods (Shao et al., 2023; Lin et al., 2024), codebook-based works (Egiazarian et al., 2024; Tseng et al., 2024), and even the deployment of fine-tuning (Liao & Monz, 2024) or QAT (Chen et al., 2024; Liu et al., 2023b) can be deployed for **MC**, additional evidences are shown in Appendix A.3.

### 3.2 PRE-LOADING MIXED-PRECISION QUANTIZATION

As outlined in Sec. 3.1, the primary storage overhead of MoE-LLMs resides in the experts, necessitating compression before loading onto devices. Mainstream LLM pruning suffers from performance degradation under extreme pruning conditions ($\geq 50\%$) (Frantar & Alistarh, 2023; Sun et al., 2023; Lu et al., 2024), whereas quantization has been demonstrated to achieve high levels of compression with lower performance drop (Huang et al., 2024b; Zhou et al., 2024). Moreover, as shown in (Li et al., 2024) and our Sec. 4.1, uniform bit-width quantization does not meet the extreme compression accuracy requirements for MoE-LLMs. Therefore, the diverse and uneven features of experts inspire us to explore the optimal mixed-precision quantization approaches.

In this section, we introduce our *Pre-Loading Mixed-Precision Quantization (PMQ)* method, designed to effectively reduce the model size by applying targeted-experts quantization. The core focus of PMQ is optimizing the bit-width allocation strategy for experts. To this end, we begin by conducting a thorough analysis of experts' behavior on the calibration dataset and leverage this information to

Figure 3: Distribution of expert drop F-norm (red), activated weights (green) and frequencies (blue) in the Mixtral $8 \times 7b$ model, encompassing 32 MoE layers with 8 experts per layer. The top set of the heatmap is calculated through C4 dataset, and the bottom set is calculated through MATH dataset. MoE-LLMs selectively activate top-2 experts in each MoE layer, wherein a significant portion of experts remain less important or inactivated all the time.

design an Integer Programming (IP) model that solves the optimal quantization configuration. For other components of the model, such as attention parameters, we apply the same bit-width.

### 3.2.1 EXPERTS SIGNIFICANCE ANALYSIS

The core principle of our expert quantization strategy is grounded in the significance of each expert, which enables the allocation of bit-widths according to their relative importance within a block. We initially observed the performance of different experts in Mixtral $8 \times 7b$ in terms of expert-drop reconstruction loss (Frobenius norm) (He et al., 2017), and activation features on the dataset C4 (Raffel et al., 2020) and the specialized domain dataset Math (Hendrycks et al., 2021). As shown in Fig 3, the impact of experts on the model varies widely: 1) some experts, such as the one at position $[2, 4]$ (Fig 3 left), have minimal influence on the output activation reconstruction loss, while others in layer 2 exhibit significantly higher losses, highlighting the imbalance among MoE-LLMs' experts; 2) the activation scores and frequencies reveal distinct patterns, where experts at positions $[11, 3]$ and $[12, 7]$ show extremely low activation frequencies and average scores, while the expert at position $[1, 3]$ has low scores but comparatively high activation frequencies; and 3) in task-specific contexts like mathematics, MoE activates fewer experts, resulting in a sparser distribution than general tasks. This variability in routing feature inspires the need to consider multiple factors in determining the optimal bit-width allocation for experts.

We mainly measure the significance of each expert through two factors: access frequency and activation weight. Given an $N$-sized calibration dataset C4 (general language understanding dataset), we first perform inference on the original 16-bit MoE-LLMs. For each expert, access frequency refers to the rate at which the expert is activated. Thus, $i$-th expert's access frequency is $\phi_i = \frac{n_i}{N}$, where $n_i$ is this expert's total activated number. A higher activation frequency indicates that the expert is triggered more often, suggesting its generality and applicability across a wide range of tokens. However, access frequency alone overlooks the potential significance of experts who are rarely activated. To account for this, we introduce the activation-weighted metric, which sums the routing weights assigned to each expert during inference. This metric for $i$-th expert can be denoted as $w_i = \frac{\sum_{j=1}^{N} \sigma_j}{N}$, where $\sigma_j$ is the expert's routing weight in the $j$-th inference. This provides a finer-grained measure of an expert's contribution in MoE-LLMs, capturing its relative importance beyond mere frequencies. The final expert significance is computed as $\phi_i^\alpha \cdot w_i^\beta$, where $\alpha$ and $\beta$ are hyperparameters used to balance the two factors.

### 3.2.2 OPTIMAL EXPERTS BIT-WIDTH ALLOCATION WITH WEIGHTED IMPORTANCE FACTORS

After obtaining the expert significance, we proceed to explore how to leverage this significance for mixed-precision quantization. The core idea is to assign different bit-widths to each expert based on its importance, preserving the contributions of more significant experts with higher bit-widths while applying more aggressive quantization to less significant experts. In addition to considering expert significance, we evaluate the reconstruction error of output activations in each MoE layer post-quantization, which allows us to quantify the impact of individually quantizing each expert. Specifically, for a given expert, we compute the Frobenius norm (F-norm) between the output of the model when this expert is quantized and the output when no quantization is applied to any experts.

$$\epsilon_{i,j} = \|\mathscr{F}(\theta) - \mathscr{F}(\theta[e_i \rightarrow Q(e_i, j)])\|_F, \tag{3}$$

where $\mathscr{F}(\theta)$ is the model output with full parameters $\theta$, and $\mathscr{F}(\theta[e_i \rightarrow Q(e_i, j)])$ represents the output when only expert $e_i$ is quantized to $j$ bits. $Q(\cdot)$ denotes the quantization function.

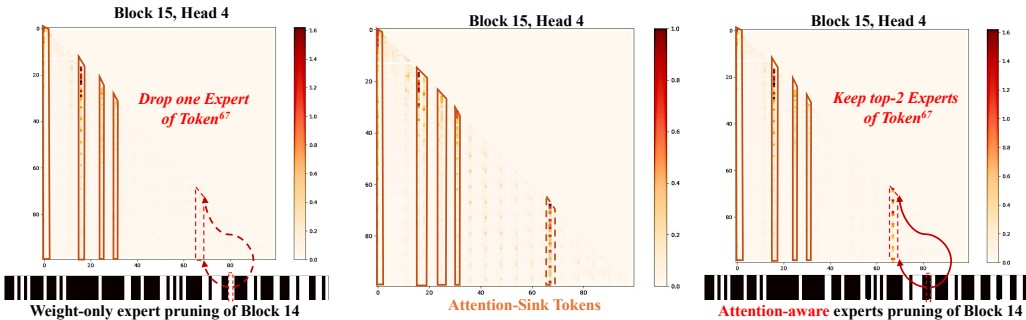

Figure 4: Typical attention map of block 15, head 4 in Mixtral $8 \times 7$b under different dynamic pruning process. The middle with out pruning shows that attention, in a column-wise manner, highlighted several tokens with high scores, such as token 31 and token 67. However, after undergoing traditional weight-only pruning through block 14 layers, experts pruned at the position of token 67, resulting in a decay in the attention map. Through attention-aware pruning based on token importance, block 14 protected token 67, thereby avoiding attention decay in the subsequent layer.

Our goal is to ensure that the extremely-low average bit-width across all experts in a MoE block equals a targeted value $k$, with bit-width options restricted to $\{1, 2, 3\}$-bit. To achieve this, we formulate the problem as an Integer Programming (IP) optimization, which only takes a second to finish the bit-width allocation computing. The IP model is defined as:

$$\text{MINIMIZE} \quad \sum_{i=1}^{n} \sum_{j=1}^{3} \phi_i^{\alpha} \cdot w_i^{\beta} \cdot (\epsilon_{i,j} \cdot x_{ij})^{\gamma}$$

$$\text{SUBJECT TO} \quad \sum_{i=1}^{n} \sum_{j=1}^{3} j \cdot x_{ij} = n \cdot k, \quad \sum_{j=1}^{3} x_{ij} = 1, \quad \forall i, \tag{4}$$

$$\sum_{i=1}^{n} x_{i3} \geq 1, \quad \sum_{i=1}^{n} x_{i2} \geq 1, \quad x_{ij} \in \{0, 1\}, \quad \forall i, j.$$

Here, $x_{ij}$ is a binary variable indicating whether the $i$-th expert is quantized to $j$ bits ($x_{ij} = 1$ if true, $x_{ij} = 0$ otherwise). To preserve the accuracy of key experts, we enforce a constraint that at least one expert must be quantized to 3 bits and at least one expert to 2 bits. $\gamma$ is a weighting hyperparameter. After determining the optimal bit-width combination for each MoE Block expert, we apply the GPTQ quantization algorithm to quantize the experts accordingly. For the remaining weights in the attention or gating module, considering their extremely small parameter size, we quantize them to 4-bit, resulting in an introduced average bit-width of no more than 0.05 bits.

## 3.3 ONLINE DYNAMIC PRUNING

Our *PMQ* strategy compresses the storage memory of experts during the pre-loading phase; however, the selection of the *top-k* experts during inference still incurs high computational costs. As discussed in Lu et al. (2024), not all tokens require $k$ experts for inference. To optimize efficiency while maintaining performance during inference, in this section, we introduce the *Online Dynamic Pruning (ODP)* technique, which identifies important tokens to retain and dynamically selects activated experts for other tokens.

### 3.3.1 ATTENTION DECAY UNDER WEIGHT-ONLY PRUNING

To effectively perform dynamic experts pruning, an intuitive and efficient method involves utilizing the *top-k* experts' routing scores during inference (Lu et al., 2024; Huang et al., 2023). This approach directly skips experts with lower routing weights among selected set for each token. For simplicity, when $k = 2$ (as in Mixtral $8 \times 7$b), the pruning process follows:

$$\{w_0 = 0, w_1 = 1, w_0, w_1 \in Top\text{-}2\{\text{G}(\mathbf{t})\} \mid \frac{w_1}{w_0} < \mu\} \tag{5}$$

$w_0$ and $w_1$ denote the *top-2* experts, respectively, with $\mu$ erving as a hyperparameter threshold for each MoE layer. This threshold is set at the median value of $\frac{w_1}{w_0}$ derived from calibration data

(Lu et al., 2024). According to Eq. (5), when a selected expert has a notably low weight, it is feasible to be pruned for the current token, thus retaining only the primary expert for computation. Sec. 4.2 documents that employing this weight-based dynamic pruning strategy reduces computational demands by 15%, but also incurs a performance decrease of approximately 10%. Further examination reveals that this decline is due to an "attention decay" effect, which is evident in Fig. 4. Specifically, unpruned conditions show a pronounced vertical pattern in the attention map of block 15, head 4, at token 67 (Fig. 4, middle). However, the application of weight-only pruning in block 14 results in the elimination of one expert for token 67, leading to a significant reduction in the attention map score at token 67 in block 15 (Fig. 4, left). This effect is herein defined as "attention decay."

### 3.3.2 SIGNIFICANCE-AWARE TOKEN PROTECTION

Weight-only pruning considers only the routing weights of experts, but overlooks the intrinsic importance of tokens. However, the output capabilities of LLMs are often influenced by a few critical tokens (Zhang et al., 2024; Guo et al., 2024; Nrusimha et al., 2024), and considering weights alone, as illustrated in Fig. 4, can lead to the pruning of experts corresponding to salient tokens. To circumvent the issue of attention decay, we introduce a simple but effective method that safeguards the computational experts of the most critical tokens in dynamic inputs from being pruned. Inspired by (Guo et al., 2024), we introduce an evaluation metric for token importance:

$$I_j = \|\mathbf{t}_j\|_1 \cdot \frac{\sum_{j \le i \le L} \mathbf{A}_{j,i}}{L - j} \tag{6}$$

where $I_j$ denotes the importance of the $i$-$th$ token, $\|\cdot\|_1$ is the $\ell_1$ norm, and the total length of input tokens is $L$. $\mathbf{A}$ represents the attention map in an LLM block, calculated from $\mathbf{A} = \text{softmax}(\frac{K^\top Q}{\sqrt{d_k}})$ of this layer. Considering the co-effects of token magnitude and attention socres, Eq. 6 flexibly combines these two factors to accurately define the importance of each token.

As demonstrated in Fig. 4, right, introducing important token protection into weight-only pruning effectively mitigates attention decay issues. Due to the high importance parameter $I_{67}$ of token 67, all experts are preserved for computing token 67 in block 14, thereby preserving the expected distribution in the attention map of block 15 for token 67. Experiments in Sec. 4.2 indicate that selectively protecting merely 2% of important tokens can significantly reduce performance losses in MoE-LLMs, while still maintaining a computational efficiency improvement of approximately 15%. We also provide the detailed computation overhead analysis in Appendix A.9.

## 4 EXPERIMENT

In this section, a series of experiments are conducted to evaluate the proposed **MC**. We present by describing the experimental parameter settings and results. In Sec. 4.1, we assess the parameter settings for the PMQ method and the performance of mixture quantization. We conduct a detailed evaluation of the performance loss and compression efficiency of ODP stage, shown in Sec. 4.2. Finally, we present the combined performance of MoE mixture compressor.

**Experiment Setup.** The mixed-precision factors of experts are calibrated from C4 (Raffel et al., 2020) dataset, with 128 sets of random sequences, each 2048 tokens long. After determining the bit-width configuration, the final quantization process follows the GPTQ (Frantar et al., 2022) procedure. We select the open-source Mixtral $8 \times 7b$ and Mixtral $8 \times 22b$ as

Table 1: Selected MoE-LLMs and model configurations. Size: the total parameter size, Act Size: activated parameter size per-token; B: decoder block number, H: hidden dimension, E: expert number.

| Model | Size | Act Size | B | H | E |
|---|---|---|---|---|---|
| Mixtral $8 \times 7b$ | 49b | 13b | 32 | 4096 | 8 |
| Mixtral $8 \times 22b$ | 141b | 39b | 56 | 6144 | 8 |

our target models, shown in Tab. 1. Mixtral $8 \times 7b$ can be compressed on two NVIDIA A100-80GB GPUs, while Mixtral $8 \times 22b$ is completed on four NVIDIA A100-80GB GPUs.

Other None-MoE layers are set to 4-bit. Due to the significant size of expert weights, the 4-bit quantization of other parameters has minimal impact on the average bit-width. In the performance experiments for the proposed **MC**, perplexity (PPL↓) was chosen as the metric to evaluate token prediction capabilities, primarily deploying the general text dataset WikiText2. To comprehensively assess the language capabilities of the compressed LLMs, we evaluated the models' overall abilities in eight zero-shot benchmarks (↑) tested by EleutherAI LM Harness (Gao et al., 2013).

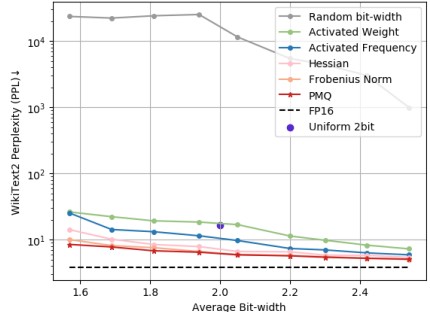

Figure 5: Quantized PPL performance of Mixtral $8 \times 7b$ under different mixed-precision strategies (with random allocation)

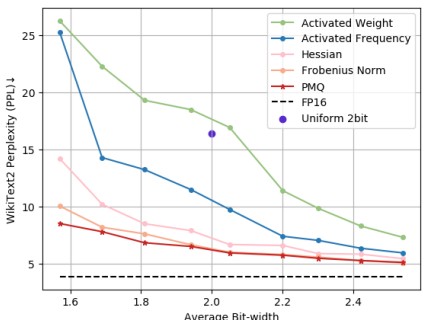

Figure 6: Quantized PPL performance of Mixtral $8 \times 7b$ under different mixed-precision strategies (without random allocation).

Table 2: Performance of quantized Mixtral $8 \times 7b$ on eight zero-shot benchmarks. We deploy GPTQ as our baseline PTQ method for uniform quantization. "Uni" denotes the uniform quantization of 2-bit with GPTQ. Since the results of some data sets in the block score predictor (BSP) (Li et al., 2024) method were not reported, we resumed the relevant quantized model from the official code repository and evaluated all the results under the same settings. In BSP, 25% MoE layers are 4-bit and the left are 2-bit to achieve 2.54-bit. "HellaS." is the short format of "HellaSwag" and "Wino." denotes "Winogrande". ↓ gives the accuracy loss between quantized results and original 16-bit model.

| Method | Bits | PIQA | ARC-e | ARC-c | BoolQ | HellaS. | Wino. | MathQA | MMLU | Avg.% ↑ |
|---|---|---|---|---|---|---|---|---|---|---|
| | 16.00 | 85.20 | 84.01 | 57.17 | 85.35 | 81.48 | 75.93 | 39.29 | 67.88 | 71.29 |
| Uni | 3.00 | 82.10 | 78.58 | 55.80 | 82.94 | 79.28 | 74.19 | 39.26 | 60.58 | $69.09_{2.2\%\downarrow}$ |
| Uni | 2.00 | 61.98 | 47.20 | 25.71 | 62.39 | 41.91 | 53.22 | 22.79 | 30.36 | $42.67_{28.6\%\downarrow}$ |
| BSP (Li et al., 2024) | 2.54 | 68.23 | 54.97 | 28.38 | 68.16 | 55.61 | 62.19 | 24.07 | 27.74 | $49.07_{22.2\%\downarrow}$ |
| Hessian (Dong et al., 2020) | 2.54 | 80.21 | 76.38 | 51.20 | 81.11 | 78.05 | 72.97 | 35.27 | 56.21 | $67.18_{4.1\%\downarrow}$ |
| | 2.05 | 75.32 | 67.26 | 45.01 | 70.29 | 71.90 | 69.11 | 31.07 | 40.85 | $58.85_{12.4\%\downarrow}$ |
| | 1.57 | 65.26 | 52.12 | 21.84 | 68.21 | 52.91 | 50.32 | 24.99 | 31.58 | $45.91_{25.4\%\downarrow}$ |
| **PMQ** | 2.54 | **80.52** | **77.10** | **51.28** | **82.54** | 79.03 | 73.95 | 39.18 | 56.37 | $\mathbf{67.50}_{3.8\%\downarrow}$ |
| | 2.42 | 80.36 | 75.76 | 50.17 | 80.00 | 78.13 | 73.09 | 34.97 | 53.22 | $65.71_{5.6\%\downarrow}$ |
| | 2.30 | 83.11 | 73.59 | 47.78 | 80.83 | 76.48 | 73.14 | 33.84 | 52.54 | $64.91_{6.4\%\downarrow}$ |
| | 2.20 | 79.05 | 73.70 | 47.87 | 74.56 | 76.63 | 72.77 | 34.24 | 47.73 | $63.29_{8.0\%\downarrow}$ |
| | 2.05 | 79.16 | 73.06 | 48.38 | 80.58 | 74.95 | 71.27 | 31.79 | 46.80 | $63.25_{8.0\%\downarrow}$ |
| | 1.94 | 76.88 | 68.48 | 45.48 | 75.23 | 72.05 | 72.61 | 31.16 | 40.93 | $60.35_{10.9\%\downarrow}$ |
| | 1.81 | 76.93 | 66.67 | 43.60 | 75.50 | 70.50 | 69.85 | 28.68 | 40.71 | $59.06_{12.2\%\downarrow}$ |
| | 1.69 | 75.41 | 64.14 | 40.61 | 68.96 | 67.01 | 68.03 | 28.04 | 37.14 | $56.17_{15.1\%\downarrow}$ |
| | 1.57 | 72.42 | 62.46 | 37.88 | 73.55 | 63.17 | 66.38 | 26.80 | 32.25 | $54.49_{16.8\%\downarrow}$ |

## 4.1 EXPERIMENT ON PRE-LOADING MIXED-PRECISION QUANTIZATION

**Ablation of Bit-Width Allocating Metrics.** Fig. 5 illustrates a significant decline in model performance with random bit-width allocation. And employing only the routing scores of experts from calibration data, the curve in Fig. 6 though better than random allocation, the PPL curve is still high. However, activation frequencies, in comparison to weight, shows a better performance.

Furthermore, in conventional networks and dense LLMs, Hessian-based quantization loss is a common use for bit-width allocation (Dong et al., 2020; Huang et al., 2024c). We also utilized is as a compared metric for expert-wise bit-width allocation. Fig. 6 also contains three metric curves: Hessian, F-norm, and PMQ. F-norm and PMQ are more effective than Hessian for expert-wise bit-width allocation, exhibiting better performance under different bit-widths. When the average bit-width exceeds 2-bit, the F-norm is similar to the PPL curve of PMQ; below 2-bit, the lead of PMQ gradually widens.

**Comparison of Mixed-Precision Quantization.** We present a comprehensive comparison of the performance of *PMQ* within the ultra-low bit-width range. GPTQ was set as the baseline for uniform bit-width quantization, denoted as "Uni" in Tab. 2. We also compare it with a recent mixed-precision

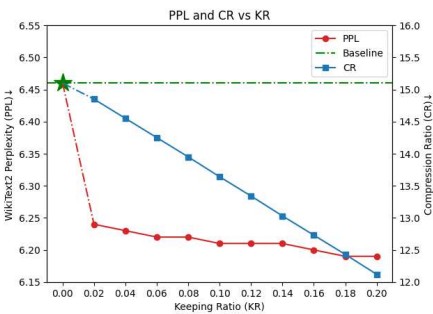
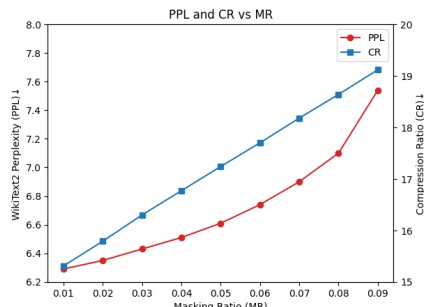

Figure 7: Significant tokens protection of 2.05-bit Mixtral $8 \times$ 7b. "CR" denotes the average computation compression ratio (blue); "PPL" denotes the perplexity(red); Star represents the weight-only pruning performance.

Figure 8: Less significant tokens drop of 2.05-bit Mixtral $8 \times$ 7b. In Mixtral $8 \times$ 7b, we mask all experts of the less significant tokens. "CR" denotes the average computation compression ratio (blue); "PPL" denotes the perplexity(red).

approach for MoE-LLMs known as the block score predictor (BSP) (Li et al., 2024). Following Eq. 4, we set the average bit-width of Mixtral $8 \times$ 7b within the range of 1.57-bit to 2.54-bit. As shown in Tab. 2, the uncompressed 16-bit model achieves an average accuracy of 71.29%. With uniform precision quantization, the average loss for the 3-bit model is approximately 2.2%, while the loss for the 2-bit model increases significantly by 28.6%, highlighting the challenges in maintaining model accuracy with existing uniform precision quantization methods at ultra-low bit widths.

BSP achieves an average accuracy of only 49.07% at 2.54-bit. However, our proposed *PMQ* achieves a performance of 67.50%, exceeding BSP by 18.4% and only falling of the 16-bit Mixtral $8 \times$ 7b by 3.8%. Notably, *PMQ* can maintain a accuracy of 54.49% at 1.57-bit, even outperforming BSP at 2.54-bit by 5.4%. The Hessian-based method in Tab. 2 consistently underperform to *PMQ* across varying bit-width levels. Specifically, Hessian shows a slight underperformance of 0.2% at 2.54-bit, while *PMQ* demonstrates more substantial advantages below 2-bit, leading by 9.4% at 1.57-bit. Furthermore, we also evaluated the few-shot capability of *PMQ* in Tab. 3, where *PMQ* continues to demonstrate superior accuracy. More bit-widths performance results are shown in Appendix A.1.

Table 3: Comparison of different mixed-precision strategies on few-shot performance (MMLU five-shot ↑) for Mixtral $8 \times$ 7b. More results of different bit-widths are shown in Appendix A.1.

| Method | Bits | Accuracy % ↑ |
|---|---|---|
| | 16.00 | 70.60 |
| Uni | 2.00 | $34.05_{36.6\%\downarrow}$ |
| BSP (Li et al., 2024) | 2.54 | $31.57_{39.0\%\downarrow}$ |
| | 2.54 | $58.22_{12.4\%\downarrow}$ |
| Hessian (Dong et al., 2020) | 2.05 | $43.51_{27.1\%\downarrow}$ |
| | 1.57 | $31.96_{38.6\%\downarrow}$ |
| | 2.54 | $\mathbf{61.19}_{9.4\%\downarrow}$ |
| **PMQ** | 2.05 | $\mathbf{49.84}_{20.8\%\downarrow}$ |
| | 1.57 | $\mathbf{33.44}_{37.2\%\downarrow}$ |

## 4.2 EXPERIMENT ON ONLINE DYNAMIC PRUNING

In pre-loading phase, *PMQ* enables the compression of MoE-LLMs to an exceptionally low bit-width range. Furthermore, during the inference phase, we apply the *ODP* outlined in Sec. 3.3 to the quantized MoE model, further enhancing the efficiency of real-time inference for lightweight models.

**Ablation of Tokens Protection.** As shown Fig. 7, when we select 2% crucial tokens to be protected, the PPL drops from 6.46 to 6.24, with activated experts' parameters decreasing only from 15.1% to 14.8%. Moreover, as we gradually increase the ratio, the performance remains relatively stable, while the compression ratio exhibits a nearly linear decline. Thus, we conclude that protecting just 2% of the important tokens can significantly enhance the compressed performance of MoE-LLMs with almost no impact on efficiency. Furthermore, we try to prune all experts associated with the less important tokens, as illustrated in Fig. 8. When removing the experts to the 2% of tokens, the overall compression ratio reached 15.8%, while the PPL improved to 6.35, yielding performance enhancements in both efficiency and accuracy compared to weight-only pruning. However, we observed that the performance curves of all experts' masking exhibited exponential growth, indicating that directly skipping experts results in a significant accuracy loss. By employing a protection mechanism for only 2% of the experts, we can maintain the accuracy of MoE-LLMs without compromising efficiency. We also provide the detailed ablation on pruning threshold in Appendix A.8.

Table 4: Ablation evaluation of *PMQ* and *ODP* for Mixtral $8 \times 7/22b$, and compared with dense LLaMA models. "Params" denotes the parameter size, and "Act Params" is averaged activated parameters for one token. The parameter calculation of the compressed model includes the compressed weights and quantizer parameters (e.g., scaling factor and zero factor for dequantization). We carry out the average activated parameter size and speedup on C4 dataset. 16-bit Mixtral $8 \times 7b$ uses 2 A100-80GB GPUs and Mixtral $8 \times 22b$ uses 4, quantized models are tested on one A100-80GB GPU.

| LLMs | Bits | PMQ | ODP | Uni | LM-Eval% ↑ | Params.(GB) | Act Params.(GB) | Speedup |
|------|------|-----|-----|-----|-----------|-------------|-----------------|---------|
| LLaMA2-7b | 16.00 | - | - | - | 61.52 | 13.48 | 13.48 | |
| LLaMA2-13b | 16.00 | - | - | - | 65.19 | 26.03 | 26.03 | |
| | 16.00 | - | - | - | 71.29 | 96.80 | 26.31 | 1.00× |
| | 2.00 | - | - | ✓ | 42.67 | 13.61 | 3.70 | 1.72× |
| | 2.54 | ✓ | - | - | 67.50 | 16.24 | 4.53 | 1.63× |
| Mixtral $8 \times 7b$ | **2.54** | ✓ | ✓ | - | **66.94** | **16.24** | **3.96** | **1.71×** |
| | 2.05 | ✓ | - | - | 63.25 | 13.41 | 3.73 | 1.67× |
| | **2.05** | ✓ | ✓ | - | **62.68** | **13.41** | **3.23** | **1.80×** |
| | 1.57 | ✓ | - | - | 54.49 | 10.82 | 2.94 | 1.82× |
| | **1.57** | ✓ | ✓ | - | **53.77** | **10.82** | **2.55** | **1.89×** |
| | 16.00 | - | - | - | 76.33 | 281.24 | 76.49 | 1.00× |
| | 2.00 | - | - | ✓ | 50.44 | 38.08 | 10.35 | 1.95× |
| | 2.54 | ✓ | - | - | 72.08 | 46.58 | 12.66 | 1.77× |
| Mixtral $8 \times 22b$ | **2.54** | ✓ | ✓ | - | **71.21** | **46.58** | **10.96** | **1.82×** |
| | 2.05 | ✓ | - | - | 67.94 | 38.35 | 10.42 | 1.80× |
| | **2.05** | ✓ | ✓ | - | **66.50** | **38.35** | **9.03** | **1.87×** |
| | 1.57 | ✓ | - | - | 59.29 | 30.27 | 8.23 | 1.97× |
| | **1.57** | ✓ | ✓ | - | **58.84** | **30.27** | **7.13** | **2.06×** |

**Memory Saving and Inference Efficiency.** Tab. 4 details the memory compression, speed tests, and average results (Gao et al., 2013) (LM-Eval) of the proposed MC. The 16-bit Mixtral $8 \times 7b$ model requires two A100-80G GPUs, while the Mixtral $8 \times 22b$ model needs four. We utilize the HQQ (Badri & Shaji, 2024) tool to save quantized weights and handle dequantization. To saving the binary weight, we design a bit-change transformation (see Appendix A.2). After applying PMQ, the Mixtral $8 \times 7b$ model can be compressed to a memory from 10.82 to 16.65 GB. During dynamic inference, ODP reduces activation parameters by about 15%, with average accuracy decreasing by less than 1%. At 2.05-bit, the average activation parameter per token is only 3.23 GB, resulting in a $1.80\times$ increase in inference speed and an evaluation accuracy of 62.68%. Tab. 4 also compares the performance of the LLaMA series dense models. The MC compressed 2.54-bit Mixtral $8 \times 7b$ model outperforms the 26.03 GB 16-bit LLaMA2-13b model, with a total parameter size of 16.65 GB and activation parameters of 3.69 GB. We have also extended the compression experiments to the Mixtral $8 \times 22b$ model. MC shows higher overall performance compared to mainstream dense models, without any training of original model. (more real speedup results are shown in Appendix A.9).

## 5 CONCLUSION

MoE represents a promising framework of sparse models for natural language understanding through scaling up the model capacity. However, the memory demands and redundancy among experts pose significant challenges for their practical implementation. In this work, we propose **MC**, a mixture compression strategy based on the imbalance of significance among experts. This method co-designs the *Pre-Loading Mixed-Precision Quantization (PMQ)* and *Online Dynamic Pruning (ODP)* approach, allowing MoE models to be compressed to an ultra-low bit-width while maintaining exceptional memory and parameter efficiency, as well as knowledgeable performance. And our mixed-precision strategy is orthogonal to various quantization techniques. Comprehensive experiments validate the effectiveness of our mixture compression, revealing that highly compressed MoE-LLMs can even outperform equal-size full-precision dense LLMs, thereby improving the feasibility of MoE compression. Future work will focus on adapting this strategy for multimodal applications and optimizing it for specific hardware platforms.

ACKNOWLEDGMENTS

This work has been supported in part by Hong Kong Research Grant Council - Early Career Scheme (Grant No. 27209621), General Research Fund Scheme (Grant No. 17202422, 17212923), Theme-based Research (Grant No. T45-701/22-R), the Innovation and Technology Fund (Mainland-Hong Kong Joint Funding Scheme, MHP/053/21), and the Shenzhen-Hong Kong-Macau Technology Research Program (SGDX20210823103537034). This research is also supported in part by National Key R&D Program of China (2022ZD0115502), National Natural Science Foundation of China (NO. 62461160308, U23B2010), "Pioneer" and "Leading Goose" R&D Program of Zhejiang (No. 2024C01161).

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

# A    APPENDIX

## A.1    MORE QUANTIZED RESULTS OF PMQ

This section expands on the comparative results of the Hessian and *PMQ* mixed precision metrics across different bit-width settings. Tab. 5 serves as an extension of Tab. 2, specifically providing a detailed comparison of the evaluation results across eight zero-shot datasets using the Hessian metric employed by HAWQ V2 (Dong et al., 2020) in the 1.57 to 2.54-bit range. Within the target bit-width interval, *PMQ* outperforms Hessian in all ranges, achieving better bit-width allocation results by 0.3% to 8.6%. Notably, at the ultra-low bit-width of 1.57-bit, *PMQ* achieves a comprehensive score of 54.49%, while Hessian reaches only 45.91%.

Table 5: Performance of quantized Mixtral $8 \times 7$b on eight zero-shot benchmarks. "HellaS." is the short format of "HellaSwag" and "Wino." denotes "Winogrande".

| Method | Bits | PIQA | ARC-e | ARC-c | BoolQ | HellaS. | Wino. | MathQA | MMLU | Avg.% ↑ |
|--------|------|------|-------|-------|-------|---------|-------|--------|------|---------|
| | 16.00 | 85.20 | 84.01 | 57.17 | 85.35 | 81.48 | 75.93 | 39.29 | 67.88 | 71.29 |
| | 2.54 | 80.21 | 76.38 | 51.20 | 81.11 | 78.05 | 72.97 | 35.27 | 56.21 | 67.18 |
| | 2.42 | 78.81 | 73.97 | 47.58 | 81.04 | 77.72 | 72.77 | 33.01 | 52.16 | 64.23 |
| | 2.30 | 79.21 | 72.41 | 46.70 | 79.15 | 76.38 | 71.25 | 31.97 | 50.60 | 63.47 |
| | 2.20 | 78.46 | 72.98 | 46.66 | 77.29 | 75.31 | 70.22 | 31.84 | 45.29 | 62.25 |
| Hessian | 2.05 | 75.32 | 67.26 | 45.01 | 70.29 | 71.90 | 69.11 | 31.07 | 40.85 | 58.85 |
| | 1.94 | 75.41 | 64.02 | 43.19 | 67.75 | 69.18 | 68.27 | 28.58 | 36.99 | 56.67 |
| | 1.81 | 71.96 | 60.81 | 37.72 | 68.27 | 63.29 | 65.46 | 26.27 | 32.58 | 53.30 |
| | 1.69 | 69.88 | 60.37 | 35.64 | 70.06 | 59.60 | 58.43 | 26.05 | 32.11 | 51.39 |
| | 1.57 | 65.26 | 52.12 | 21.84 | 68.21 | 52.91 | 50.32 | 24.99 | 31.58 | 45.91 |
| | 2.54 | **80.52** | **77.10** | **51.28** | **82.54** | **79.03** | **73.95** | **39.18** | **56.37** | **67.50** |
| | 2.42 | **80.36** | **75.76** | **50.17** | **80.00** | **78.13** | **73.09** | **34.97** | **53.22** | **65.71** |
| | 2.30 | **83.11** | **73.59** | **47.78** | **80.83** | **76.48** | **73.14** | **33.84** | **52.54** | **64.91** |
| | 2.20 | **79.05** | **73.70** | **47.87** | **74.56** | **76.63** | **72.77** | **34.24** | **47.73** | **63.29** |
| **PMQ** | 2.05 | **79.16** | **73.06** | **48.38** | **80.58** | **74.95** | **71.27** | **31.79** | **46.80** | **63.25** |
| | 1.94 | **76.88** | **68.48** | **45.48** | **75.23** | **72.05** | **72.61** | **31.16** | **40.93** | **60.35** |
| | 1.81 | **76.93** | **66.67** | **43.60** | **75.50** | **70.50** | **69.85** | **28.68** | **40.71** | **59.06** |
| | 1.69 | **75.41** | **64.14** | **40.61** | **68.96** | **67.01** | **68.03** | **28.04** | **37.14** | **56.17** |
| | 1.57 | **72.42** | **62.46** | **37.88** | **73.55** | **63.17** | **66.38** | **26.80** | **32.25** | **54.49** |

Additionally, in Tab. 6, we extend the comparison of Hessian and *PMQ*'s few-shot performance across different bit widths presented in Tab. 3. At 1.69-bit, *PMQ* achieves a score of 38.35% on the MMLU (five-shot) benchmark, while maintaining a model size that is 16% smaller than the 2-bit model under uniform quantization, with an accuracy improvement of 4.5%. More importantly, we observe that *PMQ* at 2.54-bit compresses the model size by 84% compared to the 16-bit model, yet the few-shot performance is only 9.4% lower, highlighting the substantial advantages of mixed compression for MoE models. In comparison to Hessian at the same bit-width, *PMQ* demonstrates great overall improved accuracy. BSP, on the other hand, exhibits poor performance in the few-shot evaluations, which is even lower than 2-bit uniform quantization. In Tab. 7, we also compare these precision allocating metrics on WikiText2 dataset; *PMQ* shows a clearer advantage, particularly at 1.57-bit, where it achieves a PPL of 8.50, representing a significant improvement over the 2-bit uniform quantization, while the Hessian at 1.57-bit achieves only 14.20.

## A.2    ONE-BIT WEIGHT SAVING AND DEQUANTIZATION

This paper presents MC, which explores static compression strategies and dynamic pruning method for MoE-LLMs in the ultra-low bit-width range, with selected static bit-width of 1-bit, 2-bit, and 3-bit. We observe that both 2-bit and 3-bit can be addressed using conventional linear quantizers, a method commonly utilized in most studies (Frantar et al., 2022; Shao et al., 2023; Huang et al., 2024c; Lin et al., 2024). In contrast, the quantization of 1-bit weights involves totally different calculations; we first provide the binarization formula for the weights:

$$\mathbf{B} = \mathrm{sign}(\mathbf{W}) \tag{7}$$

Table 6: Comparison of different mixed-precision strategies on few-shot performance (MMLU five-shot ↑) for Mixtral $8 \times 7b$.

| Method | Bits | Accuracy % ↑ |
|---|---|---|
| | 16.00 | 70.60 |
| Uni | 2.00 | $34.05_{28.4\%\downarrow}$ |
| BSP | 2.54 | $31.57_{30.9\%\downarrow}$ |
| Hessian | 2.54 | $58.22_{4.2\%\downarrow}$ |
| | 2.42 | $54.09_{8.3\%\downarrow}$ |
| | 2.30 | $51.37_{11.1\%\downarrow}$ |
| | 2.20 | $47.01_{15.4\%\downarrow}$ |
| | 2.05 | $43.51_{18.9\%\downarrow}$ |
| | 1.94 | $38.62_{23.8\%\downarrow}$ |
| | 1.81 | $33.87_{28.9\%\downarrow}$ |
| | 1.69 | $33.04_{29.4\%\downarrow}$ |
| | 1.57 | $31.96_{30.49\%\downarrow}$ |
| PMQ | 2.54 | $\mathbf{61.19}_{1.3\%\downarrow}$ |
| | 2.42 | $\mathbf{58.30}_{4.2\%\downarrow}$ |
| | 2.30 | $\mathbf{55.08}_{7.4\%\downarrow}$ |
| | 2.20 | $\mathbf{50.70}_{11.8\%\downarrow}$ |
| | 2.05 | $\mathbf{49.84}_{12.6\%\downarrow}$ |
| | 1.94 | $\mathbf{45.98}_{16.5\%\downarrow}$ |
| | 1.81 | $\mathbf{41.67}_{20.8\%\downarrow}$ |
| | 1.69 | $\mathbf{38.35}_{24.0\%\downarrow}$ |
| | 1.57 | $\mathbf{33.44}_{29.0\%\downarrow}$ |

Table 7: Comparison of different mixed-precision strategies on PPL performance (Wiki-Text2 PPL ↓) for Mixtral $8 \times 7b$.

| Method | Bits | PPL ↓ |
|---|---|---|
| | 16.00 | 3.84 |
| Uni | 2.00 | 16.38 |
| BSP | 2.54 | 13.61 |
| Hessian | 2.54 | 5.41 |
| | 2.42 | 5.81 |
| | 2.30 | 5.86 |
| | 2.20 | 6.58 |
| | 1.05 | 6.65 |
| | 1.97 | 7.88 |
| | 1.81 | 8.45 |
| | 1.69 | 10.18 |
| | 1.57 | 14.20 |
| PMQ | 2.54 | **5.09** |
| | 2.42 | **5.25** |
| | 2.30 | **5.45** |
| | 2.20 | **5.72** |
| | 2.05 | **5.91** |
| | 1.94 | **6.49** |
| | 1.81 | **6.81** |
| | 1.69 | **7.78** |
| | 1.57 | **8.50** |

$$\text{sign}(x) = \begin{cases} 1 & \text{if } x \geq 0, \\ -1 & \text{others.} \end{cases} \tag{8}$$

where $\mathbf{W} \in \mathbb{R}^{d \times m}$ is the full precision weight and $\mathbf{B} \in \{-1, +1\}^{d \times m}$ denotes the binarized matrix. Due to the elements range of $\mathbf{B}$ being $\pm 1$, we can not directly save the one-bit value into compact memory. Hence, we propose a simple transformation for $\mathbf{B}$:

$$\widetilde{\mathbf{B}} = \frac{\text{sign}(\mathbf{W}) + 1}{2} \tag{9}$$

where $\widetilde{\mathbf{B}} \in \{0, 1\}^{d \times m}$. In this case, we can really use 1-bit memory to storage each element. During the inference stage, we need to dequantize the binary weight and operate the matrix multiplication of each input vector follows:

$$s \cdot \mathbf{x}\mathbf{B} = s\left( \sum_{j:\widetilde{\mathbf{B}}_{ij}=1}^{d} \mathbf{x}_j - \sum_{j:\widetilde{\mathbf{B}}_{ij}=0}^{d} \mathbf{x}_j \right), \text{for } i = 1, 2, ...m \tag{10}$$

where $\mathbf{x} \in \mathbb{R}^{1 \times d}$ denotes one set of input vector (token), and $s$ represents the scaling factor of each binary matrix, which is calculated from $s = \frac{\|\mathbf{W}\|_{\ell_1}}{d \times m}$ (Rastegari et al., 2016). In this binarized weight format, we can achieve computation without minimal multiplication operation. As shown in Eq. (10), the original computation requires $dm$ multiplications and $(d-1)m$ additions, resulting in a MACs consumption of $dm$ and a computational complexity of $O(m^2)$. In contrast, binary matrix operations require only $m$ multiplications and $(d-1)m$ additions, leading to a MACs consumption of just $m$ and a computational complexity of $O(m)$.

## A.3 RESULTS OF DIFFERENT QUANTIZATION TECHNIQUES

As detailed in Sec. 3.2 of the main text, *PMQ* focuses primarily on leveraging the significance differences between experts to construct an optimal mixed-precision bit-width allocation. After determining the optimal allocation, it can be combined with various quantization techniques. In this study, to efficiently validate the effect of mixed compression, we employ GPTQ (Frantar et al., 2022), an efficient training-free post-training quantization (PTQ) strategy, which completes mixed-precision quantization on the Mixtral $8 \times 7b$ model in just 90 minutes.

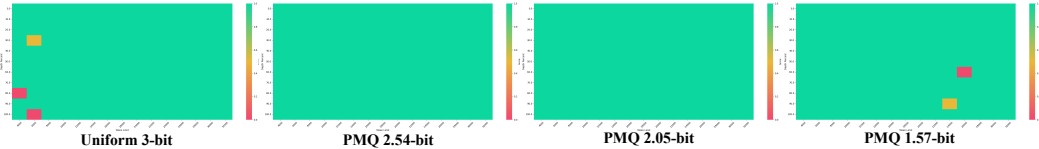

Figure 9: Needle in a Haystack evaluation. Green squares indicates a high retrieval success rate, the Y-axis represents the distance to the retrieved target.

Table 8: Performance of quantized Mixtral $8 \times 7b$ on eight zero-shot benchmarks on GPTQ (Frantar et al., 2022) and Omniquant (Shao et al., 2023). **w** denotes "with".

| Method | Bits | PIQA | ARC-e | ARC-c | BoolQ | HellaS. | Wino. | MathQA | MMLU | Avg.% ↑ |
|---|---|---|---|---|---|---|---|---|---|---|
| | 16.00 | 85.20 | 84.01 | 57.17 | 85.35 | 81.48 | 75.93 | 39.29 | 67.88 | 71.29 |
| PMQ | 2.54 | 80.52 | 77.10 | 51.28 | 82.54 | 79.03 | 73.95 | 39.18 | 56.37 | 67.50 |
| **w** *GPTQ* | 2.05 | 79.16 | 73.06 | 48.38 | 80.58 | 74.95 | 71.27 | 31.79 | 46.80 | 63.25 |
| (Frantar et al., 2022) | 1.57 | 72.42 | 62.46 | 37.88 | 73.55 | 63.17 | 66.38 | 26.80 | 32.25 | 54.49 |
| PMQ | 2.54 | 81.63 | 78.66 | 52.91 | 82.54 | 80.17 | 74.51 | 39.20 | 59.83 | 68.80 |
| **w** *Omniquant* | 2.05 | 79.77 | 74.24 | 48.65 | 81.09 | 75.76 | 72.48 | 33.01 | 47.15 | 64.01 |
| (Shao et al., 2023) | 1.57 | 73.33 | 65.28 | 38.54 | 74.06 | 66.61 | 66.59 | 26.74 | 35.20 | 55.79 |

In this section, we replace GPTQ with another advanced quantization method, Omniquant (Shao et al., 2023), which uses a learnable weight clipping (LWC) for quantization calibration. For calibration, 256 sequences from the C4 dataset are selected for gradient optimization. Omniquant requires approximately 480 minutes to quantize the Mixtral $8 \times 7b$ model (see Tab. 8), but it outperforms GPTQ across eight zero-shot benchmarks, owing to its precise search for quantizer factors via LWC. This further demonstrates the flexibility of our *PMQ* framework.

## A.4 QUANTIZATION RESULTS ON CHALLENGING BENCHMARKS

In this section, we expand our mixed-precision benchmarks on more challenging datasets in Tab. 9, considering the importance of performance testing on more complex long text or reasoning tasks (Cobbe et al., 2021; Bai et al., 2023; Chen et al., 2021). We have observed that in challenging tasks like GSM8K, HumanEval, and long-context Needle-in-a-haystack, the performance drop of model compression becomes more pronounced. This phenomenon holds true in other MoE LLM compression methods (Frantar et al., 2022; Huang et al., 2024c; Shao et al., 2023; Lu et al., 2024) as well. However, our *PMQ* method, compared to the latest method like BSP (Li et al., 2024) and HAWQ (Dong et al., 2020) with Hessian-based approaches for MoE LLM, is still able to maintain state-of-the-art performance. Fig. 9 shows the NIAH results in different sequences.

Recent studies on quantization performance losses (Jin et al., 2024; Liu et al., 2023a) were also explored, revealing that ARC-C and GSM8K primarily involves inference issues, categorized as chain-of-thought (CoT), while MMLU can be classified as in-context learning (ICL). CoT tasks, due to their intricate reasoning demands, pose significant challenges to various LLM types. Given that many open-source MoE LLMs and dense LLMs do not exhibit strong inference capabilities during pre-training, we anticipate larger performance losses when reducing model bit-width to ultra-low scenarios. The results in Tab. 2 and Tab. 9 also indicate that there is the huge potential for future exploration of MoE LLM compression on complex tasks.

## A.5 DETAILED RESULTS ON BIT-WIDTH ALLOCATION

In this section, we further visualize the different bit-width allocation results of *PMQ* on Mixtral $8 \times 7b$ model, as shown in Fig. 10. The results clearly show that the importance of MoE expert varies with different position. It can be seen that at lower bits-width, our algorithm only selects a small part of the position for protection, which greatly improves calculation efficiency. With the increasing of the bit-width, the important positions from lower bit-width are leavening unchanged which further proves the effectiveness of the proposed method.

Table 9: Comparison of different mixed-precision quantization methods on challenging benchmarks. NIAH denotes the task in Needle-in-a-haystack, which is a more challenging task for evaluating long-context ability.

| Method | Bits | GSM8K↑ | HumanEval (pass@10)↑ | NIAH↑ |
|---|---|---|---|---|
| | 16.00 | 58.30 | 59.15 | 100.00 |
| Uniform | 3.00 | 38.13 | 29.88 | 98.48 |
| Uniform | 2.00 | 0.00 | 0.00 | 0.00 |
| BSP | 2.54 | 4.25 | 3.21 | 42.21 |
| Hessian | 2.54 | 33.59 | 25.49 | 100.00 |
| Hessian | 2.05 | 17.24 | 7.84 | 93.45 |
| **PMQ** | 2.54 | **37.67** | **29.34** | **100.00** |
| **PMQ+ODP** | 2.54 | **35.25** | **27.58** | **100.00** |
| **PMQ** | 2.05 | **19.97** | **11.83** | **100.00** |
| **PMQ+ODP** | 2.05 | **18.04** | **10.02** | **99.26** |

## A.6 ABLATION ANALYSIS ON HYPER-PARAMETERS OF EXPERT SIGNIFICANCE WEIGHT

In this section, we conduct experiments based on different hyperparameter settings for the expert significance factor weights, *i.e.*, $\alpha$ and $\beta$ in Eq. 4. We evaluate these factors with values of 1, 1.5, and 2 to differentiate their relative significance on Mixtral $8 \times 7$B (2 bit). Since quantization error is a critical evaluation metric, we fix its weight $\gamma$ at 2 and vary the weights of the expert significance factors accordingly. The experimental results, shown in Tab. 10, indicate that the overall accuracy remains stable, but exhibits a slight decline when the combined value of $\alpha$ and $\beta$ exceeds the quantization error weight.

Table 10: Ablation analysis on Mixtral 7×8B model, evaluating different settings for the weights of the two significance factors, $\alpha$ and $\beta$ (Eq. 4), with the quantization error weight fixed at 2, using the WikiText2 dataset.

| | $\alpha = 1$ | | | $\alpha = 1.5$ | | | $\alpha = 2$ | | |
|---|---|---|---|---|---|---|---|---|---|
| $\beta$ | 1 | 1.5 | 2 | 1 | 1.5 | 2 | 1 | 1.5 | 2 |
| PPL | 5.92 | 5.92 | **5.91** | 5.92 | **5.91** | 5.96 | **5.91** | 5.96 | 5.95 |

## A.7 COMPARISON OF DIFFERENT TOKEN-DEPENDENT PRUNING METRIC

Regarding the dynamic pruning of experts, we note that most existing pruning methods for LLMs or other neural networks focus on static weight pruning (Sun et al., 2023; Zhang et al., 2023), and cannot dynamically prune experts during inference based on tokens. Dynamic pruning during inference remains under-explored, with only one recent post-training MoE LLM dynamic pruning work (Lu et al., 2024) proposing a gating-score-based strategy for dynamic pruning. This work has already compared with Wanda (Sun et al., 2023) method, a highly effective static pruning method, and concluded that static pruning methods result in significant performance degradation when applied to dynamic MoE LLM experts. We incorporated additional metrics for dynamic expert pruning to expand the scope of our experiments. Specifically, we perform token-dependent expert pruning on token kurtosis, token var, and token mean, where 30% of tokens will be pruned from *top*-2 to *top*-1 (Tab. 11).

## A.8 ABLATION OF DYNAMIC EXPERT PRUNING THRESHOLD

We follow the setting from recent dynamic MoE pruning work (Lu et al., 2024), selecting it as the median value of $\frac{w_1}{w_0}$, which also theoretically and empirically demonstrates that this choice of

Table 11: Comparison of different token-dependent dynamic expert pruning strategies on Mixtral $8 \times 7b$. Avg.CP denotes the average compressed parameters ratio for each token. NIAH denotes the task in Needle-in-a-haystack, which is a more challenging task for evaluating long-context ability.

| Method | $\mu(w_1/w_0)$ | Avg.CP | WikiText2↓ | LM-Eval%↑ | GSM8K↑ | HumanEval(pass@10)↑ | NIAH↑ |
|---|---|---|---|---|---|---|---|
| Token kurtosis | 0.3 | 15.62% | 7.16 | 57.22 | 14.05 | 6.54 | 93.16 |
| Token variance | 0.3 | 15.62% | 6.69 | 60.02 | 17.33 | 7.92 | 95.37 |
| Token mean | 0.3 | 15.62% | 6.82 | 59.27 | 17.76 | 6.02 | 95.65 |
| **ODP** | - | 14.88% | **6.22** | **63.25** | **18.04** | **10.02** | **99.26** |

threshold is a comprehensive optimal setting. In this section, we provide a more comprehensive ablation on threshold $\mu$ in Eq. 5. As demonstrated in Table 12, utilizing a manual threshold of 0.4, the PPL performance stands at 6.29 with a mere 12.00% of experts pruned. In contrast, our proposed method, referred to as *ODP*, achieves a PPL of 6.22 and prunes 14.88% of the experts. This not only showcases superior accuracy but also highlights enhanced efficiency.

Table 12: Ablation of different threshold hyperparameter.

| $\mu(w_1/w_0)$ | PPL (WikiText2)↓ | Avg. Pruning Params. |
|---|---|---|
| 0.4 | 6.29 | 12.00% |
| 0.5 | 6.49 | 16.51% |
| 0.6 | 6.64 | 19.25% |
| 0.7 | 6.89 | 22.43% |
| Median | 6.48 | 15.18% |
| **ODP(Median + Protection)** | **6.22** | 14.88% |

### A.9 COMPUTATION ANALYSIS OF ONLINE DYNAMIC PRUNING

During the *ODP* phase, compared to the significant reduction in the number of tokens and experts leading to large-scale matrix multiplications, the computational cost of token importance calculation can be negligible. Specifically, in Mixtral $8 \times 7b$, where the typical input token matrix size is $R^{n \times m}$, the token importance calculation involves three steps: summing attention weights, computing the $\ell_1$ norm, and performing $top$-$k$ calculations. The overall floating-point operations per second (FLOPs) calculation amounts to $n^2 + n + mn + n log n$. In the *ODP* inference phase, after dynamic pruning, an average of 15% of tokens in a MoE layer will reduce an experts inference (see Tab. 12). The FLOPs for these 15% of tokens within an expert (an expert with 3 linear layers, the size is $R^{m \times m_1}$, $R^{m_1 \times m_1}$, $R^{m_1 \times m}$) are $0.15n * \times (m \times m_1 \times 2 + m_1^2 \times 2 + m_1 \times m \times 2)$, where $m_1$ is typically much larger than n and m in Mixtral $8 \times 7b$. Therefore, the computational cost of importance calculation is usually low. As demonstrated in Tab. 4, when *PMQ* is combined with *ODP*, it further enhances computational efficiency. This indicates that the efficiency gain from experts' dynamic pruning outweighs the computational cost of token importance calculation.

Table 13: End-to-end latency comparison between FP16 and MC on Mixtral $8 \times 7b$ under different [batch, input token length]. Each cell is the latency for one token generation speed (second).

| | Hardware | [1,512] | [1,1024] | [1,2048] | [1,4096] | [8,2048] | [8,4096] | [16,2048] | [16,4096] |
|---|---|---|---|---|---|---|---|---|---|
| FP16 | $2 \times$A100 | 0.029 | 0.038 | 0.043 | 0.057 | 0.009 | 0.011 | 0.007 | 0.010 |
| MC 2.54-bit | $1 \times$A100 | 0.015 | 0.018 | 0.019 | 0.025 | 0.004 | 0.005 | 0.004 | 0.004 |
| Speedup(%) | - | 48.3 | 52.7 | 56.2 | 56.1 | 55.4 | 54.3 | 47.6 | 60.1 |

In Tab.13 and Tab.14, we present the actual speed enhancements of deployment achieved by our MC method on various hardware platforms. The speed enhancements in MC, as detailed in Tab. 13, originate from static compression during the *PMQ* phase and adaptations of the CUDA kernel (based on HQQ) along with *ODP*. Across varying batch sizes and input sequence lengths, our speedup

Table 14: Latency comparison of MoE and dense LLM under different hardware platform.

| Model | Hardware | Loading Memory | Peak GPU Memory | LM-Eval%↑ | Token/s |
|---|---|---|---|---|---|
| Mixtral 8×7b | 2×A100 | 96.8 GB | 112.6 GB | 71.29 | 23 |
| Mixtral 8×7b | 1×3090 | OOM | OOM | - | - |
| LLaMA2-13b | 1×A100 | 26.0 GB | 33.4 GB | 65.19 | 46 |
| LLaMA2-13b | 1×3090 | OOM | OOM | - | - |
| Mixtral 8×7b **MC 2.54-bit** | 1×A100 | 16.2 GB | 20.7 GB | 66.94 | 38 |
| Mixtral 8×7b **MC 2.54-bit** | 1×3090 | 16.2 GB | 20.7 GB | 66.90 | 52 |

ranges from 40% to 60%. The performance boost from model weight compression remains consistent regardless of input sequence length and batch size. However, with a fixed batch size, we notice a more pronounced speed advantage for our MC-MoE as the sequence length increases, attributed to the increased efficiency demonstrated by *ODP*. As batch size increases, both the FP16 models and compressed models experience an overall increase in throughput, leading to accelerated average token generation speeds. In Tab. 14, with MC-MoE on the RTX 3090 GPU, extreme compression allows for an average speed of 52 token/s, which is very cost-effective. In this scenario, compressed MoE LLM outperforms dense LLM in memory, accuracy, and speed.

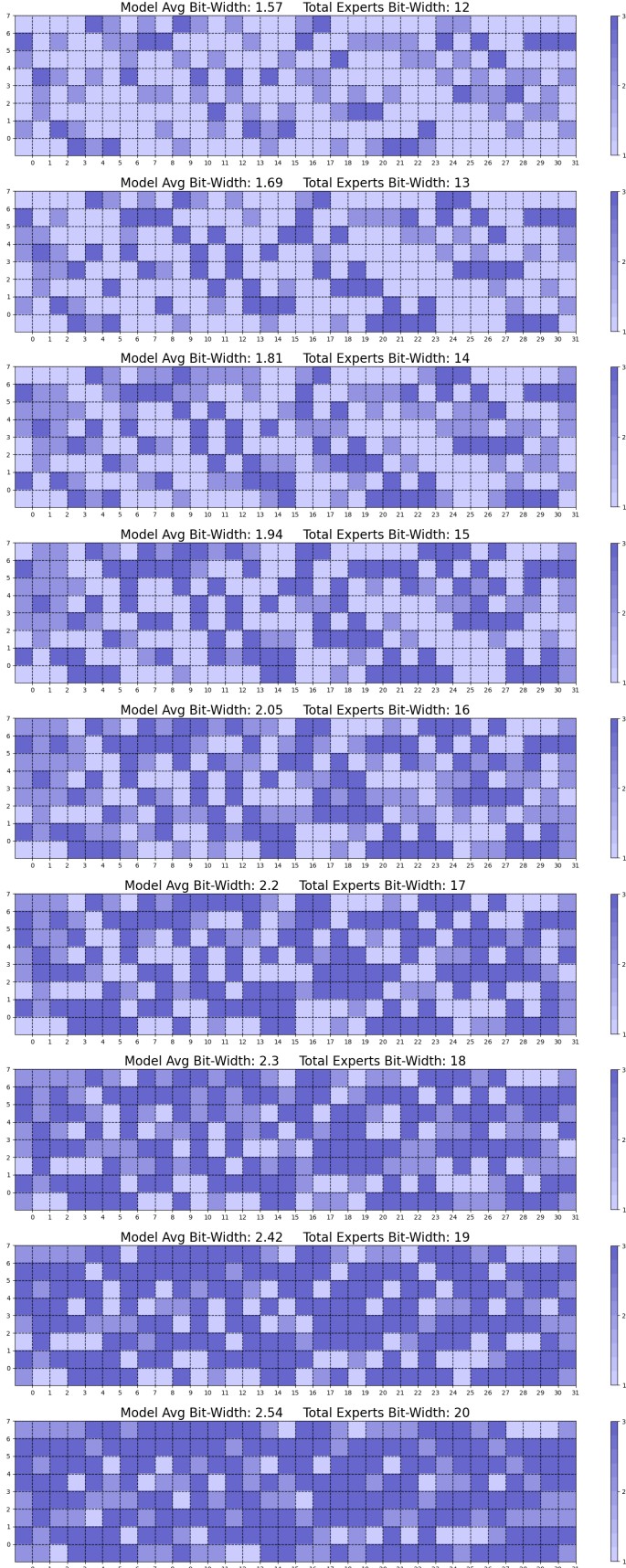

Figure 10: Visualization on different bit-width allocation. Color refers to the bit size.

