# OpenReview forum: "Mixture Compressor for Mixture-of-Experts LLMs Gains More"
_ICLR.cc/2025/Conference — ICLR 2025 Poster_

### Official Review · Reviewer_uP5F · 2024-10-27

**Soundness:** 3
**Presentation:** 3
**Contribution:** 2
**Rating:** 5
**Confidence:** 4

**Summary:**

The paper presents MC-MoE, a training-free Mixture-Compressor for Mixture-of-Experts LLMs, which is designed to address memory consumption and redundancy challenges. MC-MoE combines Pre-Loading Mixed-Precision Quantization and Online Dynamic Pruning to compress MoE-LLMs. Wxperiments show that MC-MoE maintains high performance.

**Strengths:**

Conducts lots of experiments to verify the accuracy drop of the proposed method

**Weaknesses:**

My major conerns are the motivation and practical benefit of the proposed method:

1. What is the "metric" for measuring the speedup in Table 4? Is it latency or throughput? What is the input length and output length in your benchmark setting?

2. What is the target platform of the proposed method? Cloud or edge? The main advantage of SMoE models over dense models is that they scale better in term of computation [1]. Under the same "act." param. size, SMoE is better than the dense counterpart (but non-act. experts still take GPU HBM to store). Thus it is very important to carefully design the distributed inference infrastructure to translate the theoretical benefits of SMoE to real system benefts. For example in [1], it discussed the expert load balance, pipeline parallelism, and large batch size to keep GPU busy. I did not see the discussion of the side effects of mixed precision, since it will worse the load balance, especially when the number of experts is large. If your targeted use case is on the edge, what benefits does an SMoE model offer over a dense model?

3. In practical applications, it's extremely challenging to keep GPUs fully utilized. The techniques discussed in this paper—weight-only quantization and dynamic pruning—introduce irregular computation patterns. These irregularities lead to load imbalances across GPU cores, making it difficult to maintain GPU utility. Moreover, these methods tend to perform worse with larger batch sizes. However, using large batch sizes is essential in practice to achieve optimal performance with SMoE models [1]. Therefore, while these techniques aim to improve efficiency, they may actually hinder performance due to the introduced irregularities and their incompatibility with large batch sizes required by SMoE.


[1] Outrageously Large Neural Networks: The Sparsely-Gated Mixture-of-Experts Layer

**Questions:**

See Weaknesses

---

> ### Author Response · Authors · 2024-11-22
> **Rebuttal by Authors (Part-1)**
>
> Dear Reviewer uP5F,
>
> Thank you for taking the time to provide your valuable and professional suggestions on our paper. We will address each of your questions one by one.
>
>
>
>
> > Q1: What is the "metric" for measuring the speedup in Table 4? Is it latency or throughput? What is the input length and output length in your benchmark setting?
>
> In our study, we have closely followed the methodology as described in previous studies [2] and have utilized the standard speedup testing protocols typically employed for LLMs [3]. Both references [2] and [3] utilized a consistent evaluation approach, where the performance metric involved measuring the time required by the LLM to **generate 512 tokens while maintaining the same input sequence length**, thus representing the token generation speed. We will detail this particular setup in Section 4.2 of our revised paper. Regarding the input length parameter, we have strictly adhered to the specifications provided on **page 7, line 365**, fixing the input length at **2048** tokens.
>
> [2] Not all experts are equal: Efficient expert pruning and skipping for mixture-of-experts large language models. ACL 2024
>
> [3] https://github.com/ AutoGPTQ/AutoGPTQ/
>
> > Q2: What is the target platform of the proposed method? Cloud or edge? The main advantage of SMoE models over dense models is that they scale better in term of computation [1]. Under the same "act." param. size, SMoE is better than the dense counterpart (but non-act. experts still take GPU HBM to store). Thus it is very important to carefully design the distributed inference infrastructure to translate the theoretical benefits of SMoE to real system benefts. For example in [1], it discussed the expert load balance, pipeline parallelism, and large batch size to keep GPU busy. I did not see the discussion of the side effects of mixed precision, since it will worse the load balance, especially when the number of experts is large. If your targeted use case is on the edge, what benefits does an SMoE model offer over a dense model?
>
>
>
> We appreciate your insightful question. As detailed in **Section 1** of our paper, our deployment target centers on **hardware-constrained environments**. The objective behind employing ultra-low bit-width compression techniques is to facilitate the successful deployment of MoE LLMs on **consumer-grade single GPUs or edge computing setups**, which constitute our intended platform. The rationale for selecting MoE LLMs stems from our research findings, illustrated in **Figure 1 and Table 4**, which demonstrate that compressing MoE LLMs results in performance enhancements **surpassing those of equivalently total parameters (not only activated parameters) or even larger dense LLMs**. This discovery is particularly exciting to us, suggesting that MoE LLMs may harbor greater compression potential compared to their dense counterparts. Our proposed MC-MoE compression framework enables MoE models to uphold highly competitive LLM performance levels at compression ratios nearing 80-90%, even outperforming dense LLMs of comparable sizes. We believe this breakthrough expands the horizons for leveraging MoE LLMs in edge computing scenarios.
>
> Addressing your reference to [1], we acknowledge the work's observation regarding the central challenge in MoE inference, emphasizing that "larger parameter size requires more GPUs to fit, and multi-GPU inference technology is not designed to work with MoE-based models.". Consequently, our study **is oriented towards achieving cost-effective deployment in edge computing environments through aggressive compression of model size**. The considerations you highlighted, such as expert load balancing, pipeline parallelism, and large batch sizes discussed in [1], predominantly pertain to the training phase of MoE models. As delineated in Section 1 of our paper, our initiative and objective are centered on post-training compression of MoE LLMs to enable their practical utilization in edge computing contexts.

---

> > ### Comment · Reviewer_uP5F · 2024-11-22
> > **Follow up**
> >
> > If you are under hardware-constrained environment, why choose MOE? MOE is a technique to optmize the trade-off between FLOP/s and model accuracy. Usually speaking, MOE requires more tailored system design for obtaining real benefits (like expert parallelism) due to the all-to-all communication pattern [4].
> >
> > So I am really curious about what is the benefit of using MOE under hardware-constrained environment. Can you show the apple-to-apple comparision of detailed latency (ms) between your compressed MOE model and a dense model to justify the motivation of this work (with controled input and output length)?
> >
> > Also, regarding serving MOEs, expert load balancing and expert parallelism are all needed to translate into real benefits.

---

> > > ### Author Response · Authors · 2024-11-23
> > > **Reply to Reviewer**
> > >
> > > Dear reviewer uP5F:
> > >
> > > Thank you for your timely reply! We greatly appreciate your feedback! We would like to emphasize that our primary focus is on the ability of MoE LLM to maintain superior language capabilities **under extreme compression(total memory)**, especially with our MC-MoE technology. This highlights our exploration of MoE's ultra-low bit-width compression and its applications in resource-constrained scenarios.  We have also made great efforts to include additional deployment latency experiments in the table below to further illustrate the advantages of MoE LLM in ultra-low bit-width compression.
> > >
> > > Notably, when MoE LLM (Mixtral 8\*7b, total parameter size **96.8 GB**) is compressed to 2.54 bits, the total parameter size of Mixtral 8*7b becomes **16.24 GB**, with approximately **3.96 GB** of activated parameters, achieving a performance of **66.94** in overall LLM evaluation metrics. On the other hand, the total parameter size of the mainstream open-source dense LLM model llama2-13B is **26.03 GB**, with activated parameters also at **26.03 GB**, yet it achieves a performance score of **65.19** across eight zero-shot benchmarks. This suggests that we can deploy MoE LLM with MC-MoE on consumer-grade GPUs, surpassing the performance of the larger dense LLM llama2-13B model, which **cannot be deployed** on such consumer-grade GPUs due to its size. This is truly exciting for us because in **previous compression efforts for dense LLMs**, it is **so hard to compress a dense LLM to the size of a small dense LLM while maintaining accuracy comparable to the smaller one without any additional training**. However, MoE LLM has made this achievable. More importantly, the compressed MoE LLM achieves significantly faster inference speeds (**38 token/s**) compared to the original FP16 format of MoE LLM (**23 token/s**) and is comparable with dense LLM (**46 token/s**).
> > >
> > > With MC-MoE on the RTX 3090 GPU, extreme compression allows for an average speed of **52 token/s**, which is super cost-effective. In this scenario, the compressed MoE LLM **outperforms dense LLM in memory, accuracy, and speed**. Our mixture compression for MoE LLM has also been evaluated and tested by **industry company**, proving its ability to balance total memory and accuracy. Deploying the compressed MoE LLM in a single GPU scenario can achieve better performance than dense LLM and is more cost-effective.
> > >
> > > We fully agree with what you mentioned: "MOE requires more tailored system design for obtaining real benefits (like expert parallelism) due to the all-to-all communication pattern [4],". This aligns with the core focus of many system optimization efforts for large-scale MoE models. However, our work aims to explore another advantage of MoE, the trade-off between **total memory and model accuracy (memory being crucial for resource-constrained hardware)**, allowing LLM to be more widely applicable in single-GPU and edge scenarios under less memory conditions. We believe MoE LLM has significant potential in this regard and that under such compression conditions, it may inspire more system-level optimizations for single-GPU-oriented MoE LLM, especially in expert parallelism and load balancing, as you mentioned. We believe this is a promising direction for the community!
> > >
> > > | Model                                             | Hardware              | Loading Memory | Peak GPU Memory | LM-Eval% ↑ | Token/s |
> > > | ------------------------------------------------- | --------------------- | -------------- | --------------- | ---------- | ------- |
> > > | Mixtral 8\*7b                                     | Two A100 (160 GB)     | 96.8 GB        | 112.6GB         | 71.29      | 23      |
> > > | Mixtral 8\*7b                                     | Single 3090 (24 GB)   | OOM            | OOM             | -          | -       |
> > > | LLaMA2-13b                                        | Single A100 （80 GB） | 26.0 GB        | 33.4GB          | 65.19      | 46      |
> > > | LLaMA2-13b                                        | Single 3090 (24 GB)   | OOM            | OOM             | -          | -       |
> > > | Mixtral 8\*7b (MC-MoE-2.54-bit + dynamic pruning) | Single A100 (80 GB)   | **16.2 GB**    | 20.7GB          | **66.94**  | 38      |
> > > | Mixtral 8\*7b (MC-MoE-2.54-bit + dynamic pruning) | Single 3090 (24 GB)   | **16.2 GB**    | 20.3GB          | **66.90**  | **52**  |

---

> ### Author Response · Authors · 2024-11-22
> **Rebuttal by Authors (Part-2)**
>
> > Q3: In practical applications, it's extremely challenging to keep GPUs fully utilized. The techniques discussed in this paper—weight-only quantization and dynamic pruning—introduce irregular computation patterns. These irregularities lead to load imbalances across GPU cores, making it difficult to maintain GPU utility. Moreover, these methods tend to perform worse with larger batch sizes. However, using large batch sizes is essential in practice to achieve optimal performance with SMoE models [1]. Therefore, while these techniques aim to improve efficiency, they may actually hinder performance due to the introduced irregularities and their incompatibility with large batch sizes required by SMoE.
>
> We wholeheartedly acknowledge the significance you attributed to maximizing GPU utilization. Nevertheless, our primary focus is on r**esource-constrained edge environments**, where the principal challenge lies in deploying extensive **MoE LLMs** on edge or consumer-grade GPUs. As highlighted in [4], during the inference phase, MoE models encounter two major hurdles: "the **overall model size** and the attainable memory bandwidth." Our research is geared towards mitigating the former challenge through model compression. Regarding the batch size aspect you raised, we recognize that large batch inputs typically feature prominently during the pre-training phase [1] [4]. Conversely, applications on consumer-level GPUs, particularly in edge and resource-constrained settings, often necessitate processing **smaller batch data** [4].
>
> We also concur with your observations regarding weight-only quantization and dynamic pruning as atypical computational techniques, particularly in cloud and distributed contexts. However, in the context of **edge or single GPUs**, as evidenced by our experiments, the amalgamation of these methods enables us to compress MoE LLMs by over 80% while upholding exceptional performance levels. This approach also nearly **doubles the inference speed** on a single GPU, underscoring the viability of deploying MoE LLMs in resource-constrained environments through our MC-MoE framework. Encouragingly, recent optimization endeavors in MoE systems have delved into successful quantization and dynamic expert selection [5] [6] [7]. This suggests that quantized experts retain potential for computational optimization at the system level, while dynamically selected experts offer opportunities for throughput enhancements. These commendable system optimization efforts are orthogonal to our compression algorithm work, indicating that the hybrid compression methodology we employ harbors promise for streamlined computation in both cloud and edge environments.
>
> We truly hope that our response addresses your concerns, and we welcome the opportunity to delve further into our motivations.
>
> [4] Deepspeed-moe: Advancing mixture-of-experts inference and training to power next-generation AI scale. ICML 2022.
>
> [5] Toward Efficient Inference for Mixture of Experts. NeurIPS 2024.
>
> [6] Fast Inference of Mixture-of-Experts Language Models with Offloading. Technical report 2023.
>
> [7] EdgeMoE: Fast On-Device Inference of MoE-based Large Language Models. Arxiv preprint.

---

> ### Author Response · Authors · 2024-11-25
> **Looking forward to your further reply**
>
> Dear Reviewer uP5F,
>
> We sincerely thank you for your efforts in reviewing our paper and your suggestions for enhancing this work. As we are approaching the end of the discussion period, we would like to ask whether there are any remaining concerns regarding our paper or our response. We are happy to answer any further questions.
>
> Best regards,
>
> Authors of Paper 1819

---

> > ### Comment · Reviewer_uP5F · 2024-11-25
> > **Curious about the number and implementation**
> >
> > In single GPU setting, how did you serve MoE model? Do you use a "for loop" to iterate over all tokens for the same expert, then reshape to the original input?
> >
> > I am questioning the motivation for using the MoE model in a resource-constrained environment. Generally, the running speed of an MoE model is approximately equivalent to that of a dense model that is twice its act. size—*but only if you have good system support* (like Mistral-8x7B versus Llama-2-13B), but at the cost of much larger model size. From your  profiling results, Mixtral 8*7b	 is even twice slower than a Llama-2-13B and much larger than it. Then why would I want to use it in the resource-constrained environment?
> >
> > Also, comparing the accuracy of different models is irrelevant in the context of compression work. These models are trained on different datasets, so accuracy comparisons are not meaningful. For instance, if I were to use LLaMA-3.1-8B, it would likely outperform Llama-2-13B.

---

> ### Author Response · Authors · 2024-11-26
> **Reply to Reviewer**
>
> Thanks for your reply and your time.
>
> We claim that in single GPU computation, MoE LLM **does not require** the use of 'for loop' to iterate through all tokens of the same expert. Specifically, we first compress all model weights from fp16 type to int type (averaging 2.54 bits) using PMQ technique, and through compact structured bit filling, weights require smaller total memory to store, as demonstrated in the compression of the mixture compressor to **16.2GB (from the original size of 96.8GB)** as shown in the table above. During the inference phase, we load the compressed weights into the GPU,  with input a set of tokens $X \in R^{n\times m}$, where $n$ is the total token length. Through the MoE layer, the expert-centered gating calculates the best experts for each token in a top-2 fashion. At this point, the ODP technique can **prune unimportant allocation connections**, thus reducing the number of tokens allocated to certain experts. For each $expert_i$, the tokens to be computed will become $X_i \in R^{n_i\times m}, n_i < n$, where $X_i$ denotes the assigned tokens to each expert, and therefore, the inference process of each expert on the GPU is represented as $E_i(X_i)$. Within $E_i(·)$​, parallel computation of this set of tokens is performed through matrix multiplication and addition. We accelerate the process of matrix operations on quantized matrices and input tokens using a cuda kernel design based on HQQ (https://github.com/mobiusml/hqq/tree/master/hqq/kernels). Then all the tokens are integrated together (this token allocation and reshape process fully follows the official inference code of Mistral-8x7B, lines 605-670 in https://github.com/huggingface/transformers/blob/main/src/transformers/models/mixtral/modeling_mixtral.py)) Hence, **there is no need to loop through each token**; only looping through the computation results of the experts is required (which is only 8 in each MoE layer of Mistral-8x7B, the length of token is usually thousands or tens of thousands).
>
>
> We **emphasize the meaningfulness** of using MoE LLM in resource-constrained environments because the expert mechanism of MoE LLM can enhance the learning ability of neural networks, especially in general language understanding, where this sparse learning mechanism can boost the natural language processing capability of MoE LLM. **However, as mentioned in [4] [5] [6] [7] [8] [9], a major limitation of MoE is the large model size, which is even more pronounced in LLM**. As demonstrated in our rebuttal experiments, Mistral-8x7B requires at least two A100 GPUs in terms of model size, and the inference speed may be lower than that of dense LLM models. This is primarily due to the significant I/O latency caused by the large parameters and need multi-GPU inference, **hindering the broader application of MoE LLM**. The mixture compressor we propose aims to address this issue, enabling efficient inference of MoE LLM on a single GPU, or even on consumer-grade GPUs, leading to **significant economic benefits**. Importantly, we have achieved state-of-the-art accuracy in MoE LLM compression work. While our innovative compression method, the mixture compressor, has been introduced, **the motivation for using MoE in resource-limited environments has been emphasized and recognized by many outstanding works [7] [9] [10].**
>
>
>
> Regarding your mention that comparing the accuracy of different models is irrelevant in the context of compression work, we acknowledge your point. That is also why, in the main text, we primarily compared the **accuracy results of different compression methods on the same MoE LLM**. As shown in Figure 1 of the main text, we presented evaluation results of some mainstream open-source dense LLM models to demonstrate that our framework's extreme compression of MoE LLM can still **maintain competitive overall performance to widely used open-sourced FP16 LLMs. Many ultra-low-bit-width compressions may not even guarantee that the model's performance is practical. Therefore, we aim to indirectly emphasize the practicality of our compression method.**
>
> [4] Deepspeed-moe: Advancing mixture-of-experts inference and training to power next-generation AI scale. ICML 2022.
>
> [5] Toward Efficient Inference for Mixture of Experts. NeurIPS 2024.
>
> [6] Fast Inference of Mixture-of-Experts Language Models with Offloading. Technical report 2023.
>
> [7] EdgeMoE: Fast On-Device Inference of MoE-based Large Language Models.
>
> [8] Mixtral of Experts. Technical report 2024.
>
> [9] Merge, Then Compress: Demystify Efficient SMoE with Hints from Its Routing Policy. ICLR 2024.
>
> [10] Not All Experts are Equal: Efficient Expert Pruning and Skipping for Mixture-of-Experts Large Language Models. ACL 2024

---

### Official Review · Reviewer_Z53p · 2024-11-04

**Soundness:** 3
**Presentation:** 3
**Contribution:** 3
**Rating:** 6
**Confidence:** 4

**Summary:**

This paper introduces MC-MoE, a novel compression framework for Mixture-of-Experts (MoE) Large Language Models that combines static mixed-precision quantization with dynamic pruning. The approach consists of two main components: Pre-Loading Mixed-Precision Quantization (PMQ), which allocates different bit-widths to experts based on their importance, and Online Dynamic Pruning (ODP), which identifies and protects critical tokens while dynamically pruning less important experts during inference. Through extensive experiments, the authors demonstrate that their method can greatly compress the model without too much accuracy loss.

**Strengths:**

1. The proposed PMQ method innovatively considers multiple factors (activation reconstruction error, routing scores, and frequencies) in determining bit-width allocation.
2. The paper presents a comprehensive solution that addresses both static model compression and dynamic inference optimization.
3. The authors provide extensive empirical validation across multiple benchmarks and model sizes, demonstrating the method’s robustness and scalability.

**Weaknesses:**

1. The paper lacks ablation studies on the impact of different hyperparameters (μ threshold, protection ratio) on model performance.
2. The paper does not adequately address the potential compounding effects of quantization errors across multiple MoE layers, particularly in deeper networks where error propagation could be more significant.
3. The paper lacks a comprehensive error analysis to identify which types of tasks or linguistic phenomena are most affected by the compression techniques.
4. The computational overhead of the token importance calculation in ODP is not thoroughly analyzed, which could be significant for real-time applications.

**Questions:**

1. The authors do not provide sufficient analysis of the robustness of their compression method under different input sequence lengths, which is crucial for practical deployment scenarios.
2. The generalizability of the proposed methods to other MoE architectures with different routing mechanisms is not discussed.

---

> ### Author Response · Authors · 2024-11-22
> **Rebuttal by Authors (Part-1)**
>
> Dear Reviewer Z53p,
>
> Thank you for acknowledging the innovation and contributions of our work in designing a mixed compression framework on the MoE LLM architecture and highlighting the performance breakthrough achieved by our method. We will address your questions individually and look forward to resolving any concerns you may have. We also anticipate further engaging with you in discussions.
>
> > Q1: The paper lacks ablation studies on the impact of different hyperparameters (μ threshold, protection ratio) on model performance.
>
> Firstly, we need to clarify that we conducted comprehensive ablation experiments on the protection ratio, with results presented in **Figures 7 and 8**. Detailed analysis is provided in **Section 4.2** of the paper under "**Ablation of Tokens Protection**." Specifically, **in Figure 7**, we observed the model performance and compression ratio variations by gradually increasing the protection ratio, finding that a 2% important token protection is the optimal choice. **In Figure 8**, experimenting with dropping all unimportant tokens directly across all experts showed a nonlinear increase in loss, leading us to only adopt the protection strategy in the ODP method.
>
> Regarding the definition of $\mu$, we followed the setting from recent dynamic MoE pruning work [1], selecting it as the median value of ${w_1} / {w_0}$. The authors in [1] theoretically and empirically demonstrated that this choice of $\mu$ is a comprehensive optimal setting. However, we greatly appreciate your suggestion and have included additional manual setting experiments for $\mu$ with specific results as shown in the table below:
>
> | $w_1 / w_0$ | **PPL (wikitext2)** | **Avg. Pruning Params.** |
> | ----------- | ------------------- | ------------------------ |
> | 0.4         | 6.29                | 12.00%                   |
> | 0.5         | 6.49                | 16.52%                   |
> | 0.6         | 6.64                | 19.25%                   |
> | 0.7         | 6.89                | 22.43%                   |
> | Median      | 6.48                | 15.18%                   |
> | **ODP**     | **6.22**            | 14.88%                   |
>
> Table 1: Threshold comparison experiment.
>
> [1] Not all experts are equal: Efficient expert pruning and skipping for mixture-of-experts large language models. ACL 2024
>
> > Q2: The paper does not adequately address the potential compounding effects of quantization errors across multiple MoE layers, particularly in deeper networks where error propagation could be more significant.
>
> Your inquiry regarding the accumulation of quantization errors across various layers is indeed pivotal and represents a central focus that current LLM quantization studies are predominantly addressing. The aggregation of quantization errors between blocks is responsible for performance degradation in LLMs when tackling challenging evaluation tasks, elucidating **why conventional uniform 2-bit quantization strategies struggle to uphold MoE LLM performance** (see **Table 2**). Consequently, our primary motivation revolves around precisely assessing expert imbalances, leveraging experts' activation frequency, activated weights, and the F-norm of experts' quantization losses to delineate an optimization space. This process aids in crafting the optimal mixed-precision bit-width configuration, with the goal of minimizing quantization errors as tokens traverse the MoE layer, thereby curbing error accumulation throughout different layers.
>
> As shown in **Table 2**, our mixed-precision bit-width achieved the best evaluation performance across all methods, indicating an effective reduction in the quantization loss of MoE layers. Additionally, in **Table 4**, we not only explored Mixtral 8*7B but also conducted experiments on the deeper Mixtral 8\*22B, showing that our method can effectively reduce the cumulative output quantization error of MoE layers.
>
> Furthermore, under our bit-width allocation strategy, when employing more precise quantization methods like Omniquant, our overall performance is further enhanced, surpassing even the original 3-bit model at 2.54 bits. This once again demonstrates that PMQ's mixed bit-width settings can effectively address error accumulation issues, enhancing MoE LLM performance.

---

> ### Author Response · Authors · 2024-11-22
> **Rebuttal by Authors (Part-2)**
>
> > Q3: The paper lacks a comprehensive error analysis to identify which types of tasks or linguistic phenomena are most affected by the compression techniques.
>
> Thanks. We need to clarify that our motivation is to explore the optimal mixture compression approach to ensure MoE LLM performance on general tasks with minimal model size. Exploring the impact of compression on different tasks lies beyond the scope of our current study; however, we are keen to investigate this in future research. Beyond this, we also find that our experimental results (**Table 2**) align with existing studies[2] [3], which suggest that the **chain-of-thought (CoT)** and **in-context learning (ICL)** abilities of LLMs are particularly sensitive to compression. For example, when compressing an MoE LLM to 2.54 bits, we observed a slightly greater decline in performance on ARC-C and MMLU—benchmarks within the CoT and ICL categories—compared to other tasks.
>
> This phenomenon is not unique to our work and has also been observed in other LLM compression studies [4] [5] [6]. Among the methods compared, our proposed compression strategy effectively minimizes the impact on MoE LLM performance for these benchmarks,
>
> [2] A Comprehensive Evaluation of Quantization Strategies for Large Language Models. arXiv preprint
>
> [3] Do emergent abilities exist in quantized large language models: An empirical study. ACL 2024
>
> [4] H2o: Heavy-hitter oracle for efficient generative inference of large language models. NeurIPS 2023.
>
> [5] Pruner-Zero: Evolving Symbolic Pruning Metric from scratch for Large Language Models. ICML 2024.
>
> [6] Omniquant: Omnidirectionally calibrated quantization for large language models. ICLR 2024
>
>
>
> > Q4: The computational overhead of the token importance calculation in ODP is not thoroughly analyzed, which could be significant for real-time applications.
>
> We greatly appreciate and value your comprehensive consideration of computational efficiency. We would like to clarify that during the ODP phase, compared to the significant reduction in the number of tokens and experts leading to large-scale matrix multiplications, the computational cost of token importance calculation can be negligible. Specifically, in Mixtral 8\*7B, where the typical input token matrix size is $R^{n\times m}$, the token importance calculation involves three steps: summing attention weights, computing the L1 norm of the token matrix, and performing top-k calculations. The overall FLOPs calculation amounts to $n^2 + n + m\times n + nlogn$.
>
> In the ODP calculation phase, after dynamic pruning, an average of 15% of tokens in a MoE layer will reduce experts' inference. The FLOPs for these 15% of tokens within an expert (an expert with three linear layers, the size is $R^{m\times m_1}$, $R^{m_1\times m_1}$, $R^{m_1\times m}$) are $0.15n \times (m\times m1\times 2 + m1^2\times 2 + m1\times m\times 2)$, where m1 is typically much larger than n and m in Mixtral 8*7B. Therefore, **the computational cost of importance calculation is usually low**. As demonstrated in **Table 4**, when PMQ is combined with ODP, it further **enhances computational efficiency**. This indicates that the efficiency gain from experts' dynamic pruning outweighs the computational cost of token importance calculation. We are more than willing to include this discussion on computational efficiency in the final version of the Appendix.

---

> ### Author Response · Authors · 2024-11-22
> **Rebuttal by Authors (Part-3)**
>
> > Q5: The authors do not provide sufficient analysis of the robustness of their compression method under different input sequence lengths, which is crucial for practical deployment scenarios.
>
> The experiment you suggested regarding the efficiency of sequence length compression is indeed a key in practical deployment. In our original manuscript version, we set the length to **2048** to ensure a fair comparison with other models. In reality, varying token lengths do not significantly impact the **relative efficiency** of our method. This is because the primary reduction in GPU memory usage stems from **static weight compression**, which is **independent** of input tokens. Additionally, our dynamic pruning strategy is based on the gating results for each token, and while sequence lengths may vary, the overall proportion of expert parameters pruned remains relatively consistent. Below, we present the average dynamic expert parameter pruning rates for various sequence lengths:
>
> Table 1: ODP compression ratio under different input sequence lengths.
>
> | **Sequence length** | **PPL (wikitext2)** | **Avg. Pruning Params.** |
> | ------------------- | ------------------- | ------------------------ |
> | 512                 | 7.26                | 14.92%                   |
> | 1024                | 6.94                | 15.21%                   |
> | 2048                | 6.22                | 14.88%                   |
> | 4096                | 6.18                | 14.37%                   |
>
> We have observed that as we increase the token length, the perplexity results improve. This enhancement is attributed to the utilization of longer token sequences in quantization calibration, which aids in maintaining the performance of the MoE LLM. As previously stated, the average pruning parameters during the dynamic phase remain consistent.
>
>
>
> > Q6: The generalizability of the proposed methods to other MoE architectures with different routing mechanisms is not discussed.
>
> We need to clarify that the model structure we focus on is the MoE LLM architecture [8] [9], which is also the purpose behind designing the PMQ and ODP methods. While MoE architectures come in different forms, with the most prominent being the experts-centered and token-centered structures [7], currently, nearly **all open-source MoE LLMs [8] [9] utilize the token-centered structure**. Therefore, the compression of MoE LLMs mainly targets the token-centered routing mechanism. We greatly appreciate your mention of this point, as it gives us the opportunity to further emphasize the motivation behind choosing this routing mechanism.
>
> [7] Mixture-of-Depths: Dynamically allocating compute in transformer-based language models. arXiv preprint.
>
> [8] Mixtral of experts. arXiv preprint.
>
> [9] Deepseekmoe: Towards ultimate expert specialization in mixture-of-experts language models. arXiv preprint.

---

> ### Author Response · Authors · 2024-11-25
> **Looking forward to your further reply**
>
> Dear Reviewer Z53p,
>
> We sincerely thank you for your efforts in reviewing our paper and your suggestions for enhancing this work. As we are approaching the end of the discussion period, we would like to ask whether there are any remaining concerns regarding our paper or our response. We are happy to answer any further questions.
>
> Best regards,
>
> Authors of Paper 1819

---

> > ### Comment · Reviewer_Z53p · 2024-11-25
> >
> > Thank you for the feedback. The authors have addressed most of my concerns and I will raise my score.

---

### Official Review · Reviewer_knJE · 2024-11-04

**Soundness:** 3
**Presentation:** 3
**Contribution:** 3
**Rating:** 8
**Confidence:** 4

**Summary:**

This paper proposes MC-MoE, a training-free Mixture-Compressor for MoE-LLMs that applies the significance of experts and tokens to perform deep compression, incentivized by optimization space to improve upon memory consumption of expert heads' parameters and redundancies of activated heads. To alleviate saving and loading overheads, the authors devised the Pre-Loading Mixed-Precision Quantization stage for adaptive memory allocation. Following up with Online Dynamic Pruning (ODP) stage, this method dynamically selects significant tokens to elevate inference efficiency while maintaining model performance, achieving extreme compression.

**Strengths:**

1. Pre-loading is an intuitive yet effective method to cope with overheads in loading expert parameters.
2. This method leveraged the uneven features learned by different expert heads as guidance to optimize quantization effort with integer programming while providing valid expert significance analysis to defend the assumption.
3. This method introduced token relevance from the attention heat map to the criterion of parameter pruning, offering salient pruning instructions without utilizing external clues.

**Weaknesses:**

While this research adopts weight-only pruning, we encourage the authors to compare the effectiveness of other popular pruning methods in the second stage to demonstrate the weight-only pruning is sufficient and effective among all methods selected.

**Questions:**

While the performance scores of PMQ across benchmarks were mostly robust, those of ARC-c and MMLU were relatively meager. Would the authors like to provide an in-depth analysis of these two tasks, regarding potential challenges PMQ had on particular patterns/features?

---

> ### Author Response · Authors · 2024-11-22
> **Rebuttal by Authors (Part-1)**
>
> Dear Reviewer knJE,
>
> Thank you for your enthusiastic and encouraging review of our work, which is a huge encouragement to our work! Below are our responses to each point and question raised in the review:
>
> > Q1: While this research adopts weight-only pruning, we encourage the authors to compare the effectiveness of other popular pruning methods in the second stage to demonstrate the weight-only pruning is sufficient and effective among all methods selected.
>
> We greatly appreciate your suggestion, as we believe it will further enhance the advantages of the ODP method proposed in our paper. Our thorough research into existing pruning methods revealed that most current LLM or other popular neural network pruning works mainly focus on **static weight pruning** such as [1] [2] [3], concentrating on **offline** model pruning. Unfortunately, these methods can not be used for dynamic expert pruning in MoE LLM inference.
>
> Due to the unique computational characteristics of MoE LLM, some experts are selectively activated during the inference of each token, a dynamic activation feature unique to the MoE architecture. This prompted us to delve into dynamic pruning research tailored for MoE LLM. Our dynamic expert pruning, inspired by the most recent MoE compression work [4], leverages MoE's unique gating mechanism to dynamically prune experts. Building on this, we observed the attention decay feature, as described in Figure 4. Hence, we introduced a token importance preservation mechanism that significantly improves the pruning accuracy of MoE LLM with minimal impact on efficiency.
>
> While current popular pruning methods for LLM do not support expert pruning during dynamic inference, we highly value your suggestion. Following your advice, we have expanded the experiments of dynamic expert pruning by selecting token mean, token variance, and token kurtosis as metrics for comparison in the table below:
>
> Table 1: Comparison of different token-dependent dynamic expert pruning strategies on Mixtral 8*7b.
>
> | **Metric** | **threshold** | **PPL (wikitext2)** | **Avg. Pruning Params.** |
> | ---------- | ------------- | ------------------- | ------------------------ |
> | Kurtosis   | 0.3           | 7.16                | 15.62%                   |
> | Var        | 0.3           | 6.69                | 15.62%                   |
> | Mean       | 0.3           | 6.82                | 15.62%                   |
> | **ODP**    | -             | **6.22**                | 14.88%                   |
>
>
>
> [1] A simple and effective pruning approach for large language models. ICLR 2024
>
> [2] H2o: Heavy-hitter oracle for efficient generative inference of large language models. NeurIPS 2023.
>
> [3] Pruner-Zero: Evolving Symbolic Pruning Metric from scratch for Large Language Models. ICML 2024.
>
> [4] Not All Experts are Equal: Efficient Expert Pruning and Skipping for Mixture-of-Experts Large Language Models. ACL 2024

---

> ### Author Response · Authors · 2024-11-22
> **Rebuttal by Authors (Part-2)**
>
> > Q2: While the performance scores of PMQ across benchmarks were mostly robust, those of ARC-c and MMLU were relatively meager. Would the authors like to provide an in-depth analysis of these two tasks, regarding potential challenges PMQ had on particular patterns/features?
>
> We greatly appreciate the recognition of PMQ's robust performance in various LLM evaluations. The observation you made regarding the more pronounced performance loss reduction in common-sense and other metrics like ARC-C and MMLU benchmarks during LLM compression is indeed a correct and intriguing phenomenon. In our study, when activation parameters were compressed by 85%, we saw a decrease of 6% and 9% in ARC-C and MMLU (0-shot), respectively, whereas the existing MoE LLM compression strategy BSP showed reductions of 29% and 40% in the same benchmarks.
>
> Through comprehensive comparative research, we found this phenomenon present in various prior works related to LLM compression [3] [4] [5] [6] [7], where ARC-C and MMLU show significant decreases as quantization bit-width decreases across multiple methods and other quantization works. Recent studies on quantization performance losses [6] [7] were also explored, revealing that ARC-C primarily involves inference issues, categorized as **chain-of-thought (CoT)**, while MMLU can be classified as **in-context learning (ICL)**. Detailed analyses in [7] highlighted the changes in the quantization scale for these two types of data tasks. CoT tasks, due to their intricate reasoning demands, pose significant challenges to various LLM types. Given that many open-source MoE LLMs and Dense LLMs do not exhibit strong inference capabilities during pre-training, we anticipate larger performance losses when reducing model bit-width to ultra-low scenarios. In ICL tasks, substantial performance drops may occur, mitigated by adding prompt samples to enhance ICL performance, as shown in Table 3. The conclusions in [7] indicate that performance can be significantly restored through post-quantization fine-tuning in CoT and ICL tasks, aligning with our method's compatibility with fine-tuning, as noted in Section 1.
>
> We find your insights to be professional and valuable. We plan to incorporate this discussion in the revised version's experimental analysis and cite relevant articles, enhancing the professionalism of our work.
>
> [5] Omniquant: Omnidirectionally calibrated quantization for large language models. ICLR 2024
>
> [4] A Comprehensive Evaluation of Quantization Strategies for Large Language Models. arXiv preprint
>
> [7] Do emergent abilities exist in quantized large language models: An empirical study. ACL 2024

---

### Official Review · Reviewer_vMGD · 2024-11-13

**Soundness:** 3
**Presentation:** 2
**Contribution:** 3
**Rating:** 8
**Confidence:** 4

**Summary:**

The authors proposed two designs for efficient MoE inference: PMQ and ODP. PMQ conducts mix-precision quantization of experts, and ODP essentially prunes experts depending on certain critical tokens.

**Strengths:**

- MoE efficiency is a relatively under explored area in comparison to general, dense LLMs. This work is a welcomed addition.
- Two proposed designs are able to deliver decent task performance, especially for PMQ over BSP in Table 2.

**Weaknesses:**

- The novelty of the proposed work is limited, as both mixed-precision quantization and token-dependent expert pruning are well-explored avenues for efficient MoE inference.
- Potential lack of baseline: BSP is the only truly relevant comparison to PMQ due to its mixed-precision approach. No pruning comparisons are provided for ODP.
- Most datasets used in Table 2 are common-sense intelligence tasks. Extensive literature across various fields has shown that such tasks (and ppl) are relatively robust and can achieve substantial compression gains. I would like to see more challenging tasks evaluated, such as GSM8k, HumanEval and LongBench.
- The efficiency evaluation seems somewhat rough. The registered speedup does not correspond to the featured task. I would like to see comprehensive reports on latency, throughput (across different tasks, compression rates, and batch sizes), and memory.

I also have a few formatting suggestions:
- Please consider adding highlights in Figure 3, as you have done in Figure 4. It is difficult to identify the discussion substance by reading coordinates on a small diagram.
- The LM-Eval column in Table 4 is not defined, though I understand it refers to the tasks in Table 2.
- Not that it matters much, but the authors may want to know that the proper way to use left and right quotation marks in LaTeX is `` and '', respectively. Currently, only right quotation marks are being used.

**Questions:**

-  I don't seem to find much connection between the PMQ and ODP part of the submitted work. Are they standalone to each other?

---

> ### Author Response · Authors · 2024-11-22
> **Rebuttal by Authors (Part-1)**
>
> Dear Reviewer  vMGD,
>
> Thank you for highlighting and appreciating our contributions in exploring the compression of MoE LLM! Below, we will address your questions one by one.
>
> > Q1: The novelty of the proposed work is limited, as both mixed-precision quantization and token-dependent expert pruning are well-explored avenues for efficient MoE inference.
>
> Thank you for your feedback. While there have been some explorations[1] [6] in MoE quantization and pruning, it is still in the early stages. Compression of MoE LLM is **not yet a well-solved challenge**. To our knowledge, most approaches have focused on quantization bit-width greater than 3-bit[1] [2] [7], with limited attention given to ultra-low-bit scenarios. Their performance tends to degrade significantly in low-bit-width settings (<3-bit) [1]. The current dynamic expert pruning method [6] suffers great performance degradation due to the ignoration of saliency tokens. When we want to compress the MoE LLM in both the static memory and the dynamic activated parameters phase, the current methods can not successfully meet the performance requirement for real deployment.
>
> In contrast, our work represents **one of the first comprehensive efforts** to advance MoE-LLMs into the ultra-low-bit regime (< 3-bit), aiming to deploy high accuracy, and further enable adaptive, efficient inference by addressing the inherent redundancies in MoE-LLMs. To achieve this, we developed a PMQ method for ultra-low-bit quantization and an adaptive pruning technique ODP tailored for efficient inference of MoE-LLM.
>
> The key innovations of our approach and comparisons with existing methods are summarized as follows:
>
> - (a) Although existing method, BSP[1], has explored mixed-precision quantization for MoE LLMs at the expert level, they suffer from significant performance degradation (see Table 1). And the traditional Hessian-based method [2] can not work well on MoE LLM. This is primarily due to their design, which fails to account for the unique activation characteristics of MoE LLMs, such as activation frequency and weight distribution. (b) In contrast, our proposed mixed-precision quantization method, **PMQ**, explicitly incorporates these unique characteristics by designing the awareness of experts’ activation features and formulating them into an Integer Programming optimization to determine the optimal expert-level precision configuration. This approach effectively mitigates expert activation features with quantization loss, preserving model performance. Experimental results demonstrate the advantages of PMQ, including superior low-bit performance on both PPL and 8 zero-shot benchmarks, as shown in Figure 5, Figure 6, and Table 2.
> - (a) Most research on expert pruning has focused on static weight pruning [3] [4] [5]. However, these methods, like Wanda [3], can not support the dynamic pruning-activated experts in the online inference phase. While the only existing work [6] on dynamic pruning employs a gating-score-based strategy. However, directly pruning based on gating-score can weaken the impact of important tokens in MoE LLM, creating a performance bottleneck. (b) In contrast to [6], which only considers the threshold of the gating score, we observe that this approach causes **attention decay** in subsequent layers, negatively impacting overall performance. Building on this insight, we propose a simple yet effective protection mechanism based on token importance, which safeguards only 2% of the most critical tokens, avoiding the degradation of significant tokens. This strategy successfully reduces the model’s perplexity (PPL) from 6.47 to 6.22. Despite its simplicity, we believe these findings provide valuable insights for advancing dynamic pruning techniques.
>
> In summary, compared to existing methods, our work **is the first to successfully achieve static quantization and dynamic pruning at the same time for MoE LLM compression**. Furthermore, we **achieve state-of-the-art performance** in both static and dynamic phases independently. Notably, as shown in Table 2, Figure 7, and Table 4, we deliver the best mixed-precision performance for MoE LLMs at ultra-low bit widths, effectively mitigate the performance loss from dynamic expert pruning without compromising efficiency, and are the first to achieve high-accuracy MoE LLM with a parameter compression ratio of 85.8%–90.3% and acceleration close to 2×.

---

> > ### Author Response · Authors · 2024-11-22
> > **Referenced works**
> >
> > [1] Examining Post-Training Quantization for Mixture-of-Experts: A Benchmark. arXiv preprint.
> >
> > [2] Hawq: Hessian aware quantization of neural networks with mixed-precision. ICCV 2019.
> >
> > [3] A simple and effective pruning approach for large language models. ICLR 2024.
> >
> > [4] H2o: Heavy-hitter oracle for efficient generative inference of large language models. NeurIPS 2023.
> >
> > [5] Pruner-Zero: Evolving Symbolic Pruning Metric from scratch for Large Language Models. ICML 2024.
> >
> > [6] Not All Experts are Equal: Efficient Expert Pruning and Skipping for Mixture-of-Experts Large Language Models. ACL 2024.
> >
> > [7] EdgeMoE: Fast On-Device Inference of MoE-based Large Language Models. Arxiv preprint.

---

> ### Author Response · Authors · 2024-11-22
> **Rebuttal by Authors (Part-2)**
>
> > Q2: Potential lack of baseline: BSP is the only truly relevant comparison to PMQ due to its mixed-precision approach. No pruning comparisons are provided for ODP
>
> To evaluate PMQ, we provide a comprehensive benchmark **comparing various mixed-precision quantization methods**, addressing the relatively underexplored area of quantization for MoE LLMs. Few studies, such as BSP, have been specifically tailored to this task. As shown in **Figures 5, 6, and Table 2,** our benchmark includes **five mixed-precision quantization strategies** for MoE LLMs: BSP, the Hessian-based strategy from HAWQ, Average Activated Weight, Average Activated Frequency, and Frobenius Norm-Based Quantization Loss. We believe this benchmark can provide a solid foundation for future studies.
>
> For ODP, which focuses on dynamic expert pruning during inference, the **only existing work [6]** proposes a gating-score-based strategy for dynamic expert pruning, and we include it for comparison in our paper. However, most pruning methods for LLMs target static weight pruning [3] [4] [5], which is fundamentally different from dynamic expert pruning and cannot be directly applied to solve dynamic pruning. Static weight pruning and dynamic pruning represent orthogonal strategies for improving the inference efficiency of LLMs. Exploring their potential combination could be an exciting direction for future research.
>
> Although current popular pruning methods for LLMs do not support dynamic expert pruning during inference, we highly value your suggestion. Following your advice, we incorporated additional metrics for dynamic expert pruning to expand the scope of our experiments. We perform token-dependent expert pruning on token kurtosis, token var, and token mean, where 30% of tokens will be pruned from top-2 to top1. The details are as follows:
>
>
>
> | **Metric** | **threshold** | **PPL (wikitext2)** | **Avg. Pruning Params.** |
> | ---------- | ------------- | ------------------- | ------------------------ |
> | Kurtosis   | 0.3           | 7.16                | 15.62%                   |
> | Var        | 0.3           | 6.69                | 15.62%                   |
> | Mean       | 0.3           | 6.82                | 15.62%                   |
> | **ODP**    | -             | **6.22**                | 14.88%                   |
>
> Table 1: Comparison of different token-dependent dynamic expert pruning strategies on Mixtral 8*7b.
>
> We have also included Table 1 in the Appendix.

---

> ### Author Response · Authors · 2024-11-22
> **Rebuttal by Authors (Part-3)**
>
> > Q3: Most datasets used in Table 2 are common-sense intelligence tasks. Extensive literature across various fields has shown that such tasks (and ppl) are relatively robust and can achieve substantial compression gains. I would like to see more challenging tasks evaluated, such as GSM8k, HumanEval and LongBench.
>
> We sincerely agree with and appreciate the reviewer's suggestion to extend the evaluation of compression to more challenging datasets. In the manuscript, we have selected the PPL metric on the WikiText-2 dataset, along with metrics from eight common-sense benchmarks, as the core experiments. This approach aligns with most prior compression studies in the LLM field, as well as earlier work on compressing MoE LLMs. As shown in the experimental results, our proposed method significantly outperforms existing methods on these metrics.
>
> Following your suggestion, we made our best efforts within the limited time available to add evaluation experiments on GSM8K and HumanEval for our method and the comparative strategies in the manuscript. However, due to the current limitations of LongBench, which supports only a few models and requires unique prompts for each model, we could not obtain fair results at this stage (the FP16 model’s score is also 0.00). After the double-blind review process, we hope to engage further with excellent works like LongBench to expand the evaluation of MoE LLMs and contribute more to the community. These benchmarks (GSM8K, LongBench, HumanEval) and results(GSM8K, HumanEval) will be discussed and cited in the revised version of the manuscript.
>
> Table 2: Comparison of different mixed-precision quantization methods on challenging benchmarks.
>
> | **Method** | **Bits** | **Params.** | **GSM8K** | **HumanEval (pass@10)** |
> | ---------- | -------- | ----------- | --------- | ----------------------- |
> |            | 16.00    | 96.80 GB    | 58.30     | 59.15                   |
> | Uniform    | 3.00     | 19.66 GB    | 38.13     | 29.88                   |
> | Uniform    | 2.00     | 13.61 GB    | 0.00      | 0.00                    |
> | BSP        | 2.54     | 16.24 GB    | 4.25      | 3.21                    |
> | Hessian    | 2.54     | 16.24 GB    | 33.59     | 25.49                   |
> | Hessian    | 2.05     | 13.41 GB    | 17.24     | 7.84                    |
> | **PMQ**    | 2.54     | **16.24 GB**    | **36.78**     | **29.34**                   |
> | **PMQ**    | 2.05     | **13.41 GB**    | **19.97**     | **11.83**                   |
>
> We have observed that in challenging tasks like GSM8K and HumanEval, the performance drop of model compression becomes more pronounced. This phenomenon holds true in other MoE LLM compression tasks as well. However, our PMQ method, compared to the latest methods like BSP and HAWQ with Hessian-based approaches for MoE LLM, is still able to maintain state-of-the-art performance. Notably, our approach, utilizing 2.54 bits per weight, achieves performance comparable to 3-bit models while reducing storage requirements by approximately **3.4 gigabytes**. The results in the table also indicate the potential for future exploration of the performance preservation of MoE LLM compression on complex tasks.

---

> ### Author Response · Authors · 2024-11-22
> **Rebuttal by Authors (Part-4)**
>
> > Q4: The efficiency evaluation seems somewhat rough. The registered speedup does not correspond to the featured task. I would like to see comprehensive reports on latency, throughput (across different tasks, compression rates, and batch sizes), and memory.
>
> We appreciate your suggestions, as they will help us clarify the efficiency contributions of our work. Specifically, we would like to clarify that in **Table 4** of our paper, we detailed comparisons of model **inference latency, total memory of model weights, and the total parameter activations of MoE LLM during inference under various compression ratios**. For instance, at 1.57 bits, we compressed the total weight memory of Mixtral 8 × 7b from 96.70GB to 10.82GB, with an average activation parameter of 2.55GB on the C4 dataset, resulting in a 1.89x speedup in generating 512 tokens.
>
> Regarding your suggestion about testing on different featured tasks, we did explore this in the initial experimental phase. However, we found that the **latency differences in inference across different featured tasks were not significant**. This is because our mixture compression method's speed improvement for models primarily benefits from PMQ's ultra-low bit-width compression. The speed gains from quantizing the MoE LLM weights themselves are completely independent of the task and input, aligning with the experimental trends shown in Table 4. Thus, we did not include speed test experiments on different featured tasks in the final version of the paper. We genuinely hope that this clarification of our experimental setup is helpful to you.
>
>
>
> > Q5: few formatting suggestions
>
> Thank you for taking the time to provide valuable feedback on our work. Your insights are truly appreciated and will undoubtedly help enhance the clarity and quality of our publication.
>
> We will diligently address each of the points you have raised:
>
> 1. We will add highlights to Figure 3, similar to what has been done in Figure 4, to improve the readability and comprehension of the diagram.
>
>
>
> 2. We will clearly define the LM-Eval column in Table 4, ensuring that its relationship to the tasks outlined in Table 2 is explicitly stated for better understanding.
>
>
>
> 3. Thank you for pointing out the correct usage of left and right quotation marks in LaTeX. We will make the necessary adjustments to ensure consistency and adherence to the proper formatting standards.
>
> Once again, we extend our deepest gratitude for your thorough review and constructive feedback. Your input is invaluable in refining our work to meet the highest standards of excellence.
>
>
>
> > Q6:I don't seem to find much connection between the PMQ and ODP part of the submitted work. Are they standalone to each other?
>
> Your understanding is absolutely correct. PMQ and ODP represent orthogonal directions for enhancing the efficiency of MoE LLMs from the perspectives of static memory phase and dynamic activated parameter phase. In this work, we aim to conduct a comprehensive study of MoE-based LLMs and explore novel methods that harness the unique properties of MoE to improve inference speed and reduce memory costs.
>
> Further, as simply compressing both phases can lead to a significant drop in model performance, we also introduced the MC-MoE hybrid compressor, which integrates static quantization and dynamic pruning to achieve extreme compression of MoE-LLM while minimizing accuracy loss, ensuring the optimal balance between performance and efficiency.  Extensive experiments have validated the effectiveness of our approach.

---

> > ### Comment · Reviewer_vMGD · 2024-11-23
> >
> > Thank you for the detailed rebuttal. Many of my concerns (Q1, Q3, Q5, Q6 — following your categorization) are largely resolved or alleviated. However, I have a few more requests.
> >
> > * For Q2, can you evaluate these pruning methods with real tasks, like the commonsense ones you have done, and maybe also pick the best one to undergo a full eval with the several tasks I suggested? I don't find PPL-only evaluation helpful, especially when the gaps are small.
> > * For Q3, can you also provide the same GSM8k & HumanEval evaluation with ODP alone, and PMQ + ODP?
> > * For Q3, if LongBench results are unobtainable for now, can you at least supply an NIAH test with a noisy background like Paul Graham? On the same note, many general long context benchmark works have employed these two evals rather extensively; I'd recommend checking out [1, 2] should the authors have difficulty running such evals.
> > * For Q4, can you provide throughput for different input lengths and batch sizes? Something like FastGen provided here would be ideal: https://openreview.net/forum?id=uNrFpDPMyo&noteId=E5LpHGpiYi
> >
> > ---
> >
> > [1] KV Cache Compression, But What Must We Give in Return? A Comprehensive Benchmark of Long Context Capable Approaches
> > [2] A Controlled Study on Long Context Extension and Generalization in LLMs

---

> > > ### Author Response · Authors · 2024-11-25
> > > **Reply to Reviewer (Part-1)**
> > >
> > > Dear Reviewer vMGD:
> > >
> > > Thank you for your timely reply and informative suggestion! We are so pleased that our previous rebuttal has solved most of your questions and concerns. We have been doing our best to expand the experiment in the past few dozen hours. We will reply to your comments one by one below.
> > >
> > >
> > >
> > > > Reply 1: For Q2, can you evaluate these pruning methods with real tasks, like the commonsense ones you have done, and maybe also pick the best one to undergo a full eval with the several tasks I suggested? I don't find PPL-only evaluation helpful, especially when the gaps are small.
> > >
> > > Great suggestion! Here we also evaluate these pruning metric on eight zero-shot benchmarks, and list their average accuracy as LM-Eval% ↑. We can observed that our ODP shows **leading performance** on both **PPL and CommomSense benchmarks**. We will also renew the table in our revised pdf version. Thanks again for your insightful suggestion.
> > >
> > > Table 1: Comparison of different token-dependent dynamic expert pruning strategies on Mixtral 8*7b.
> > >
> > > | **Metric** | **threshold** | **Avg. Pruning Params.** | **PPL (wikitext2)** | LM-Eval% ↑ |
> > > | ---------- | ------------- | ------------------------ | ------------------- | ---------- |
> > > | Kurtosis   | 0.3           | 15.62%                   | 7.16                | 57.22%     |
> > > | Var        | 0.3           | 15.62%                   | 6.69                | 60.02%     |
> > > | Mean       | 0.3           | 15.62%                   | 6.82                | 59.27%     |
> > > | **ODP**    | -             | 14.88%                   | 6.22                | **63.25%** |
> > >
> > >
> > >
> > > > Reply 2: For Q3, can you also provide the same GSM8k & HumanEval evaluation with ODP alone, and PMQ + ODP?
> > >
> > > Thank you for your suggestion! In the table below, we have followed your advice and provided the experimental results for PMQ + ODP on GSM8k & HumanEval. As depicted in **Figure 7**, **Figure 8** , the dynamic pruning setting is based on the static quantization of 2.05 bits (PMQ). When evaluating ODP, we aimed to align with real-world applications, meaning that dynamic pruning targets models that have already been compressed through static quantization. Therefore, the metrics mentioned in `Reply-1 (Table 1)`, as well as in Figure 7 and Figure 8, are all based on PMQ (2.05-bit). We will also include the new table in the revised appendix.
> > >
> > > Table 2: Comparison of different mixed-precision quantization methods on challenging benchmarks and longcontext test.
> > >
> > > | **Method**  | **Bits** | **Params.**  | Act Params. | **GSM8K** | **HumanEval (pass@10)** |
> > > | ----------- | -------- | ------------ | ----------- | --------- | ----------------------- |
> > > |             | 16.00    | 96.80 GB     | 26.31 GB    | 58.30     | 59.15                   |
> > > | Uniform     | 2.00     | 13.61 GB     | 3.70 GB     | 0.00      | 0.00                    |
> > > | BSP         | 2.54     | 16.24 GB     | 4.66 GB     | 4.25      | 3.21                    |
> > > | Hessian     | 2.54     | 16.24 GB     | 4.53 GB     | 33.59     | 25.49                   |
> > > | Hessian     | 2.05     | 13.41 GB     | 3.73 GB     | 17.24     | 7.84                    |
> > > | **PMQ**     | 2.54     | 16.24 GB     | 4.53 GB     | 36.78     | 29.34                   |
> > > | **PMQ+ODP** | 2.54     | **16.24 GB** | **3.96 GB** | **35.25** | **27.58**               |
> > > | **PMQ**     | 2.05     | 13.41 GB     | 3.73 GB     | 19.97     | 11.83                   |
> > > | **PMQ+ODP** | 2.05     | **13.41 GB** | **3.23 GB** | **18.04** | **10.02**               |

---

> > > ### Author Response · Authors · 2024-11-25
> > > **Reply to Reviewer (Part-2)**
> > >
> > > > Reply 3: For Q3, if LongBench results are unobtainable for now, can you at least supply an NIAH test with a noisy background like Paul Graham? On the same note, many general long context benchmark works have employed these two evals rather extensively; I'd recommend checking out [1, 2] should the authors have difficulty running such evals.
> > >
> > > Thank you for providing great works and more information on longcontext technology! Although our work isn't specifically focused on compressing and optimizing for longcontext applications, the details you gave us have allowed us to broaden our evaluation of general LLM/MoE LLM compression methods.
> > >
> > > By following your suggestion and testing NIAH (with https://github.com/jzhang38/EasyContext) on our mixture compression models within the 4k to 32k length range (Mixtral 8*7B has a max length of 32k), we evaluated the scores of various static compression methods. We found that our mixed-precision quantization compression method performed more consistently on the NIAH evaluation, achieving **100% accuracy at 2.54-bit and 2.05-bit**, even **surpassing the 3-bit results of uniform quantization**. We will also include the NIAH image results in the final revised version's Appendix.
> > >
> > > Your professional advice is greatly appreciated, and we consider this to be highly significant as, to our knowledge, this is the first evaluation of weight-only compression on longcontext NIAH, demonstrating stable performance. Evaluating longcontext provides a fresh perspective on weight-only compression.
> > >
> > > Table 3: Evaluation results on Needle-in-a-haystack benchmarks.
> > >
> > > | **Method**  | **Bits** | **Params.**  | NIAH       |
> > > | ----------- | -------- | ------------ | ---------- |
> > > |             | 16.00    | 96.80 GB     | 100.00     |
> > > | Uniform     | 3.00     | 19.66 GB     | 98.48      |
> > > | Uniform     | 2.00     | 13.61 GB     | 0.00       |
> > > | BSP         | 2.54     | 16.24 GB     | 42.21      |
> > > | Hessian     | 2.54     | 16.24 GB     | 100.00     |
> > > | Hessian     | 2.05     | 13.41 GB     | 93.45      |
> > > | **PMQ**     | 2.54     | **16.24 GB** | **100.00** |
> > > | **PMQ+ODP** | 2.54     | **16.24 GB** | **100.00** |
> > > | **PMQ**     | 2.05     | **13.41 GB** | **100.00** |
> > > | **PMQ+ODP** | 2.05     | **13.41 GB** | **99.26**  |
> > >
> > >
> > >
> > >
> > >
> > > > Reply 4: For Q4, can you provide throughput for different input lengths and batch sizes? Something like FastGen provided here would be ideal: https://openreview.net/forum?id=uNrFpDPMyo&noteId=E5LpHGpiYi
> > >
> > > This is a really helpful link! Here we provide the **[batch, input length]** experiments to evaluate the average generation speed for one token. We flexibly sliced the token length on C4 datasets and batch them into GPU, then test the total latency with second (smaller is better).
> > >
> > > Table 4: End-to-end latency comparison between FP16  and MC-MoE on Mixtral 8\*7b. The column name "Settings" represents [batch, input token length] tested. Each cell is the latency for one token generation speed (second).
> > >
> > > | Settings              | Hardware            | [1,512] | [1,1042] | [1,2048] | [1,4096] | [8, 2048] | [8,4096] | [16, 2048] | [16, 4096] |
> > > | :-------------------- | ------------------- | :------ | :------- | :------- | :------- | :-------- | -------- | ---------- | ---------- |
> > > | FP 16                 | Two A100 (160 GB)   | 0.029 s | 0.038 s  | 0.043 s  | 0.057 s  | 0.009     | 0.011    | 0.007      | 0.010      |
> > > | MC-MoE (2.54-bit+ODP) | Single A100 (80 GB) | 0.015 s | 0.018 s  | 0.019 s  | 0.025 s  | 0.004     | 0.005    | 0.004      | 0.004      |
> > > | Speed-up(%)           | -                   | 48.3%   | 52.7%    | 56.2%    | 56.1%    | 55.4%     | 54.3%    | 47.6%      | 60.1%      |
> > >
> > > The speed improvements in MC-MoE, as indicated in Table 1, stem from static compression during the **PMQ phase** and **CUDA kernel modifications (based on HQQ)** along with **ODP**. Across various batch sizes and input sequence lengths, our **speedup (%) ranges between 40% to 60%**. The speed advantage from weight compression of the model itself is independent of input sequence length and batch size. However, with a fixed batch size, we observe a **more pronounced speed advantage** for our MC-MoE as the sequence length increases, thanks to the further efficiency exhibited by **ODP technology's dynamic pruning**. As the batch size increases, both FP16 models and compressed models see an overall throughput increase, resulting in faster average token generation speeds.
> > >
> > > This analysis confirms that the MC-MoE compression method can significantly accelerate MoE LLMs in practical applications while maintaining performance in multitask evaluations. While complex parallelism and very large batch inference are not central to edge scenarios, we believe this will inspire more work in optimizing more complex parallel inference tasks based on this foundation in the future.

---

> ### Comment · Reviewer_vMGD · 2024-11-25
> **Good rebuttal. Bumping to 8, but a few more mainly cosmetic recommendations.**
>
> I appreciate the authors for fulfilling my hefty requests in a timely and comprehensive manner. As the authors may have sensed, I am very much opposed to only evaluating common-sense intelligence tasks (and, by extension, PPLs), as I do not believe these are accurate measures of whether a compressed model retains the capabilities of the full one.
>
> This is less of a concern for MoE efficiency works because most baselines still fail to maintain performance on such common-sense intelligence tasks, so achieving good accuracy retention might still be a reasonable first step. However, I strongly recommend that the authors feature more challenging tasks and highlight them for your readers — e.g., GSM8K and HumanEval, as I mentioned above; I am also almost certain that tasks like LongBench and MMLU/MMLU-Pro will present challenges — it is not wise to continue focusing on easy tasks with <10% accuracy gaps pre/post compression when there are more severe ability degradation issues that require attention. I understand that authors often hesitate to present imperfect results, but in this case, the proposed method has already demonstrated strong performance advantages over the baseline, so this should not be a concern.
>
> On the same note, please consider moderating claims like below, or at least specifying the datasets being referred to:
>
> > *"For instance, at 2.54 bits, MC-MoE compresses 76.6% of the model, with only a 3.8% average accuracy loss. During dynamic inference, we further reduce activated parameters by 15%, with a performance drop of less than 0.6%."*
>
> My final two recommendations are as follows:
>
> 1. My Q2 follow-up reads:
>
> > For Q2, can you evaluate these pruning methods with real tasks, like the commonsense ones you have done, and maybe also pick the best one to undergo a full eval **with the several tasks I suggested?**
>
> You completed the full LM-Eval, but you did not include tasks like GSM8K, HumanEval, etc. I do not believe the inclusion of these tasks will alter your conclusions given the already observed performance gap, but please fulfill these evaluations in full.
>
> 2. There appears to be another MoE efficiency work titled *Merge, Then Compress: Demystify Efficient SMoE with Hints from Its Routing Policy*, which already named their method MC-SMoE. This is quite similar to your MC-MoE name. Please consider adjusting your naming to prevent confusion for future audiences.

---

> > ### Author Response · Authors · 2024-11-25
> > **Thanks to Reviewer for raising score to 8!**
> >
> > Dear Reviewer  vMGD,
> >
> > Thank you so much for the professional feedback and suggestions. We are delighted that our responses and experiments addressed your questions, and we sincerely thank you for raising your score to 8!
> >
> > Your insights provided us with deeper steps of compression evaluation in the field of LLM and MoE. We fully agree with your professional perspective on the compression works, we should evaluate LLM capabilities on a broader range of challenging benchmarks, not only common-sense intelligence tasks. This is crucial for effectively applying LLM in real applications.
> >
> > We believe that not only in this work do we need to increase the dimensions of evaluation, but in future research, we will advocate for using GSM8K, HumanEval, NIAH, LongBench, and MMLU/MMLU-Pro, or even more comprehensive evaluations. We hope to collaborate with the community to focus on compression losses in more comprehensive and complex tasks, making compression more practical!
> >
> > We followed your suggestion and added specific benchmarks to the statements in our abstract. After organizing all supplementary tables and experiments, we supplemented all benchmark evaluation results from Q2 in the final revised article. Additionally, we have carefully considered the potential issue of name overlap with the MC-SMoE work you mentioned and directly named our proposed mixture compressor framework as MC in the revised version.
> >
> > Finally, please allow us to express our most sincere thanks to you!
> >
> > Best regards, Paper 1819 Authors

---

### Author Response · Authors · 2024-12-02
**Grateful Response to all Reviewers**

Dear Reviewers and ACs:

We would like to express our gratitude to all the reviewers for their hard work and dedication and for their contributions to the academic community during discussion phase.

As the discussion phase is coming to a close, if you have any questions or concerns regarding the article, we are more than happy to address them promptly!

Thank you once more for your support and collaboration!

Best regards,

Paper 1819 Authors

---

### Meta-Review · Area_Chair_YD2g · 2024-12-21

**Metareview:**

This paper presents MC-MoE, a compression framework for Mixture-of-Experts (MoE) LLMs that integrates Pre-Loading Mixed-Precision Quantization (PMQ) and Online Dynamic Pruning (ODP). The proposed method achieves significant compression (e.g., 76.6% reduction at 2.54 bits with a 3.8% accuracy drop) while maintaining competitive performance. Strengths include the innovative integration of static quantization and dynamic pruning, extensive empirical validation, and relevance to scalable deployments. Weaknesses include limited novelty, incomplete ablations on task-specific performance, and insufficient exploration of compounding errors across layers. The decision to accept is based on its contributions to scalable LLMs, strong empirical results, and effective rebuttals addressing reviewer concerns.

**Additional Comments On Reviewer Discussion:**

The reviewers raised concerns about the novelty, baselines, and evaluation scope of the proposed methods. During the rebuttal, the authors addressed these by conducting additional experiments, including evaluations on GSM8K and HumanEval, and providing detailed analyses of token importance calculations and the robustness of PMQ across various settings. While some minor concerns about task generalizability and error propagation in deeper networks remain, the authors’ thorough responses and added experiments have resolved most issues. This engagement reflects the authors' commitment to improving the paper and addressing critical feedback, reinforcing the decision to accept.

---

### Decision · Program_Chairs · 2025-01-22

Accept (Poster)